# Protection afforded by post-infection SARS-CoV-2 vaccine doses: A cohort study in Shanghai

Bo Zheng[1†], Bronner P Gonçalves[2†], Pengfei Deng[3†], Weibing Wang[1,4], Jie Tian[1], Xueyao Liang[1], Ye Yao[5]*, Caoyi Xue[3]*

[1]Department of Epidemiology, School of Public Health, Fudan University, Shanghai, China; [2]Department of Comparative Biomedical Sciences, School of Veterinary Medicine, University of Surrey, Guildford, United Kingdom; [3]Shanghai Pudong New Area Center for Disease Control and Prevention, Shanghai, China; [4]Key Laboratory of Public Health Safety of Ministry of Education, Fudan University, Shanghai, China; [5]School of Public Health, Shanghai Institute of Infectious Disease and Biosecurity, Fudan University, Shanghai, China

*For correspondence:
yyao@fudan.edu.cn (YY);
xuecaoyi83@163.com (CX)

[†]These authors contributed equally to this work

## eLife assessment

This **important** work by Zheng and colleagues uses a large cohort database from Shanghai to identify that post-infection vaccination among previously vaccinated individuals provides significant low to moderate protection against re-infection. The evidence supporting the conclusion is **convincing** with some limitations, e.g., lack of symptom severity as an outcome, and no inclusion of time since infection as an independent variable. This study will be of interest to vaccinologists, public health officials and clinicians.

## Abstract

**Background:** In many settings, a large fraction of the population has both been vaccinated against and infected by severe acute respiratory syndrome coronavirus 2 (SARS-CoV-2). Hence, quantifying the protection provided by post-infection vaccination has become critical for policy. We aimed to estimate the protective effect against SARS-CoV-2 reinfection of an additional vaccine dose after an initial Omicron variant infection.

**Methods:** We report a retrospective, population-based cohort study performed in Shanghai, China, using electronic databases with information on SARS-CoV-2 infections and vaccination history. We compared reinfection incidence by post-infection vaccination status in individuals initially infected during the April–May 2022 Omicron variant surge in Shanghai and who had been vaccinated before that period. Cox models were fit to estimate adjusted hazard ratios (aHRs).

**Results:** 275,896 individuals were diagnosed with real-time polymerase chain reaction-confirmed SARS-CoV-2 infection in April–May 2022; 199,312/275,896 were included in analyses on the effect of a post-infection vaccine dose. Post-infection vaccination provided protection against reinfection (aHR 0.82; 95% confidence interval 0.79–0.85). For patients who had received one, two, or three vaccine doses before their first infection, hazard ratios for the post-infection vaccination effect were 0.84 (0.76–0.93), 0.87 (0.83–0.90), and 0.96 (0.74–1.23), respectively. Post-infection vaccination within 30 and 90 days before the second Omicron wave provided different degrees of protection (in aHR): 0.51 (0.44–0.58) and 0.67 (0.61–0.74), respectively. Moreover, for all vaccine types, but to different extents, a post-infection dose given to individuals who were fully vaccinated before first infection was protective.

**Conclusions:** In previously vaccinated and infected individuals, an additional vaccine dose provided protection against Omicron variant reinfection. These observations will inform future policy decisions on COVID-19 vaccination in China and other countries.

**Funding:** This study was funded the Key Discipline Program of Pudong New Area Health System (PWZxk2022-25), the Development and Application of Intelligent Epidemic Surveillance and AI Analysis System (21002411400), the Shanghai Public Health System Construction (GWVI-11.2-XD08), the Shanghai Health Commission Key Disciplines (GWVI-11.1-02), the Shanghai Health Commission Clinical Research Program (20214Y0020), the Shanghai Natural Science Foundation (22ZR1414600), and the Shanghai Young Health Talents Program (2022YQ076).

## Introduction

Four years after the first reports of severe acute respiratory syndrome coronavirus 2 (SARS-CoV-2) infection, the Coronavirus disease (COVID-19) pandemic continues to be a global concern, especially due to the risk of emergence of new variants (*Pan et al., 2023*; *Zhu et al., 2020*). In most countries, the variant that is currently epidemiologically dominant is the Omicron (*Viana et al., 2022*; *Zhou et al., 2023*), which, due to its increased transmissibility and high number of mutations, led to significant increases in the number of infections in 2022 (*Li et al., 2021*). Omicron variant infections were first observed in China in December 2021 (*Viana et al., 2022*), and in Shanghai, the spread of the Omicron BA.2 sublineage led to a substantial increase in COVID-19 incidence between February 26 and June 30, 2022 (*Chen et al., 2022b*).

In December 2022 (*Zheng et al., 2023*), an important change in the COVID-19 policy in China, namely the end of most social distancing measures and of mass screening activities, was associated with a second surge in SARS-CoV-2 infections in Shanghai. The current circulation of the virus in the Shanghainese population and reports of vaccine fatigue mean that it is important to estimate the protective effect of vaccination against reinfection in this population. In this study, we aimed to quantify the effect of vaccine doses given after a first infection on the risk of subsequent infection. For that, we used data collected during the first Omicron variant wave, when hundreds of thousands of individuals tested real-time polymerase chain reaction (RT-PCR)-positive for SARS-CoV-2 infection (*Huang et al., 2023*) in Shanghai, of which 275,896 individuals in Pudong District. The fact that the population in Shanghai was mostly SARS-CoV-2 infection naïve before the spread of the Omicron variant provides a unique opportunity to estimate the real-world benefit of post-infection vaccine doses in a population that was first exposed to infection during a relatively short and well-defined time window. We further investigated whether the number of pre-infection vaccination doses modified the protective effect of the post-infection dose against Omicron BA.5 sublineage. To avoid ambiguity in the text, in the following sections, we often refer to vaccine doses given after the initial infection as 'post-infection vaccination' or 'post-infection vaccine dose'.

## Materials and methods
### Study setting and participants

During the first Omicron variant wave, the entire city of Shanghai entered a lockdown phase on April 1, 2022; and on June 1, 2022, the local government declared the end of the city-wide lockdown (*Chen et al., 2022b*). Residents (citizens, including immigrants from other provinces, and foreigners) in Shanghai, a provincial-level municipality in China with a population of more than 25 million people, underwent several rounds of SARS-CoV-2 RT-PCR testing between April 1 and May 31, 2022. This study included individuals diagnosed with their first SARS-CoV-2 infection between April 1 and May 31, 2022 in the Pudong District, which is a large and densely populated district of Shanghai spanning an area of 1210 square kilometers with a permanent resident population of 5.57 million, served by more than 30 hospitals and 60 community health centers (*Xinguang, 2017*); both individuals who were diagnosed by mass screening and those with symptoms who were seen by healthcare professionals were included. Information on infection history as well as data on demographic variables (sex at birth, and age) were provided by the Center for Disease Control and Prevention in Shanghai, China.

Additional information (e.g. on occupation, residence, clinical severity, and symptoms of first infection) was available for patients with hospital records and those who were transferred to a hospital.

The recorded clinical severity was categorized as asymptomatic, mild, moderate, severe, or critically ill (*Chen et al., 2022a*; *Ge et al., 2021*). During the first Omicron variant wave in Shanghai, many Fangcang shelter hospitals were rapidly converted into facilities to treat COVID-19 patients and made important contributions in providing adequate healthcare to patients with mild-to-moderate symptoms, and preventing further viral transmission in the community (*Zhang et al., 2022*). The efficient referral and transfer mechanisms in the local communities meant that the majority of patients were admitted to Fangcang shelter hospitals 1 or 2 days after testing positive for SARS-CoV-2 (*Ye et al., 2022*). However, many patients were either admitted to hospital without complete clinical information or not transferred to other hospitals; for these study participants, information on clinical severity was often missing.

### Study design and eligibility criteria

There was a second surge of Omicron variant cases from December 2022; and from January 2023, free nucleic acid testing services were no longer offered in Shanghai, and mandatory PCR testing on all personnel ceased. After this change, the cost of an individual test was 16 yuan (US$2.33; *Jian, 2023*). For this reason, the outcome variable in our analysis was based on reinfection data collected prior to January 2023. Reinfection-related death was defined as death within 30 days of a SARS-CoV-2 reinfection (*Nielsen et al., 2022*); individuals who died from reasons unrelated to COVID-19 between the two Omicron variant waves were excluded. Individuals were also excluded if the date of first infection was missing. As the reasons why some individuals refused to receive vaccination varied and were often unknown (*Wu et al., 2021*), our analysis focused on individuals who had received at least one vaccine dose before the first SARS-CoV-2 infection.

### Vaccination and reinfection data

The Shanghai Group Immunization System captures all vaccine administrations in the municipality and is updated daily. This system is linked to the National Immunization Program Information System, which also includes national identification-matched COVID-19 vaccinations received outside of Shanghai (*Huang et al., 2023*). Vaccination status was categorized in accordance with national technical recommendations for COVID-19 vaccination (*Huang et al., 2023*): unvaccinated, partially vaccinated, fully vaccinated, and fully vaccinated with booster dose (here, also referred to as booster vaccination). Here, inactivated vaccines refer to Sinovac-CoronaVac, Sinopharm/BIBP COVID-19 vaccine, and Sinopharm/WIBP COVID-19 vaccine; Ad5-vectored vaccine refers to Cansino Ad5-nCoV-S COVID-19 vaccine; and recombinant protein vaccine refers to recombinant COVID-19 vaccine (CHO cell), Anhui Zhifei Longcom Biopharmaceutical Institute of Microbiology.

As in other epidemiological studies (*National Center for Immunization and Respiratory Diseases (U.S.). Division of Viral Diseases, 2020*; *Sotoodeh Ghorbani et al., 2022*), we defined reinfection as a positive SARS-CoV-2 RT-PCR or rapid antigen test at least 90 days after the first positive test. Phylogenetic analysis coupled with contact tracing data revealed community transmission of Omicron BA.5.2 sublineage in Shanghai (*Huang et al., 2022*); the subvariant was estimated to have caused ~90% of infections during the second Omicron variant wave.

### Statistical analysis

Continuous variables are summarized as medians and interquartile ranges (IQR), and categorical variables, as counts, proportions, or percentages. Cumulative incidence of reinfections was calculated and expressed as the number of reinfected individuals per 100 participants. Cox proportional hazards models with a time varying exposure variable corresponding to post-infection vaccination were used to estimate adjusted hazard ratios (aHRs). In this analysis, time to reinfection was the outcome, and post-infection vaccination, the exposure of interest; models were adjusted for sex, age, residence, occupation, and clinical severity of the first SARS-CoV-2 infection. As, due to social distancing measures, the number of SARS-CoV-2 infections diagnosed between May and November 2022 in Shanghai was low ($N$ = 89 individuals), the start of the follow-up in the survival analysis was on December 1, and for each participant, the follow-up continued until January 3, 2023 or confirmed SARS-CoV-2 reinfection, whichever occurred earlier. We note that by using this approach, both the start of the follow-up and the eligibility (which required confirmed SARS-CoV-2 infection during the initial Omicron variant wave, and being at risk of infection when the second Omicron variant wave occurred in December)

are temporally aligned. On the other hand, for some participants, the time of exposure (post-infection vaccination) occurred after time zero, which could potentially lead to immortal time bias (*Hernán et al., 2016*); for this reason, the exposure variable was as time-varying. We also assessed whether this effect was modified by demographic characteristics.

We conducted a secondary analysis to estimate the protection afforded by post-infection vaccination before the second Omicron variant wave (i.e. only vaccine doses received before December 2022 were used in defining the exposure variable); this analysis was stratified by time intervals between post-infection vaccination and the second Omicron variant wave, and by post-infection vaccine type (for individuals who had the same vaccination status or the same number of doses before first infection). For this secondary analysis, propensity score matching was used to improve comparability between the exposed and unexposed groups; the propensity score was calculated using a logistic regression model with all available baseline characteristics, and a one-to-one matching was performed using the nearest neighbor matching method with a caliper width of 0.20 (*Ueno et al., 2021*). We assessed the balance of covariates after matching using standardized mean differences (SMDs), and considered a value of less than 0.1 to be indicative of adequate matching (*Appendix 1—tables 1–3*). All statistical analyses were performed using R.4.1.1 software (Foundation for Statistical Computing, Vienna, Austria; https://www.r-project.org). We followed the Strengthening the reporting of observational

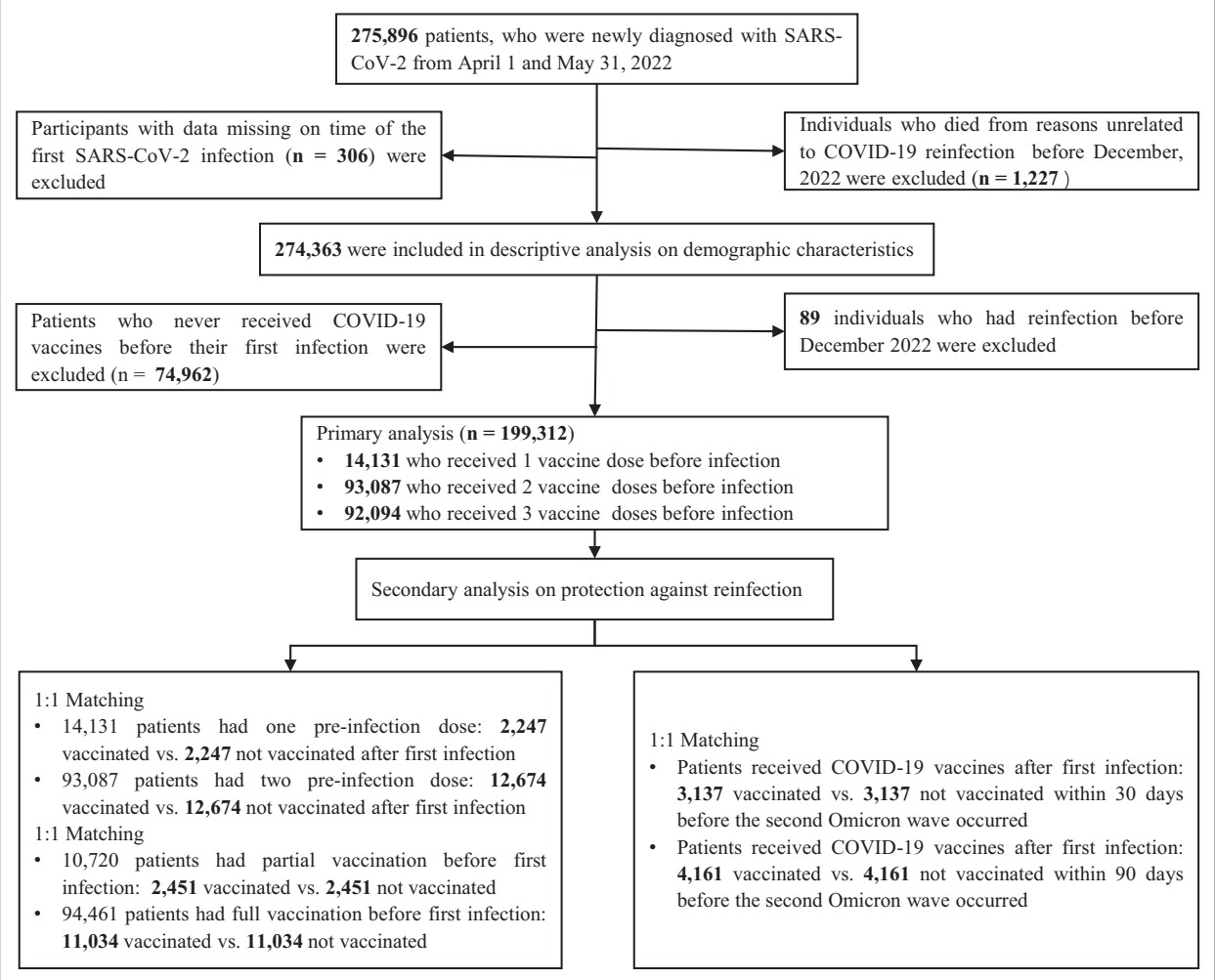

**Figure 1.** Flow chart describing the selection of participants for the analysis. The number of individuals in this figure is not the same as some of the numbers in *Table 1* because of missing data in key variables. Note that in the bottom part of the chart, related to secondary analyses, the boxes represent overlapping sets of study participants; in other words, some individuals included in the secondary analyses that correspond to the left box were also included in analyses corresponding to the box on the right.

**Table 1.** Characteristics of the study population and reinfection rate by post-infection vaccination status.

Here, reinfection rate refers to the percentage of the relevant study subpopulation with evidence of reinfection between December 1, 2022 and January 3, 2023. Note that for the variables on region, occupation, and clinical severity, data are missing for large fractions of the study population. Note also that information was only available on sex at birth, but not on gender.

| Characteristics | | All N (%) | All Reinfection rate, % (95% CI) | No post-infection vaccination N (%) | No post-infection vaccination Reinfection rate, % (95% CI) | Post-infection vaccination N (%) | Post-infection vaccination Reinfection rate, % (95% CI) |
|---|---|---|---|---|---|---|---|
| Overall | | 199,312 | 24.4 (24.2, 24.6) | 183,165 | 24.7 (24.5, 24.9) | 16,147 | 21.5 (20.8, 22.2) |
| Sex | Male | 112,672 (56.5) | 26.1 (25.8, 26.4) | 104,002 (56.8) | 26.4 (26.1, 26.7) | 8670 (53.7) | 22.3 (21.3, 23.3) |
| | Female | 85,804 (43.1) | 22.4 (22.1, 22.7) | 78,403 (42.8) | 22.9 (22.6, 23.2) | 7401 (45.8) | 17.4 (16.4, 18.3) |
| Age, years | 0–6 | 1736 (0.9) | 7.0 (5.8, 8.3) | 1569 (0.9) | 6.6 (5.4, 8.0) | 167 (1.0) | 10.2 (6.2, 15.9) |
| | 7–19 | 10,762 (5.4) | 13.0 (12.3, 13.7) | 10,347 (5.6) | 12.9 (12.3, 13.6) | 415 (2.6) | 14.7 (11.4, 18.8) |
| | 20–39 | 75,955 (38.1) | 22.4 (22.1, 22.8) | 71,005 (38.8) | 22.7 (22.3, 23.0) | 4950 (30.7) | 19.1 (17.9, 20.3) |
| | 40–59 | 74,680 (37.5) | 29.4 (29.0, 29.8) | 70,569 (38.5) | 29.6 (29.2, 30.0) | 4111 (25.5) | 25.8 (24.2, 27.4) |
| | 60+ | 35,903 (18.0) | 22.6 (22.1, 23.1) | 29,446 (16.1) | 23.7 (23.1, 24.2) | 6457 (40.0) | 17.6 (16.6, 18.6) |
| Regions | Shanghai | 44,259 (22.2) | 18.9 (18.5, 19.3) | 41,250 (22.5) | 19.4 (19.0, 19.9) | 3009 (18.6) | 11.9 (10.7, 13.1) |
| | Other provinces | 44,959 (22.6) | 20.9 (20.5, 21.3) | 43,045 (23.5) | 21.0 (20.6, 21.5) | 1914 (11.9) | 18.2 (16.4, 20.2) |
| Occupations | Preschoolers and students | 12,232 (6.1) | 12.1 (11.5, 12.7) | 11,677 (6.4) | 12.0 (11.4, 12.6) | 555 (3.4) | 13.2 (10.4, 16.4) |
| | Employed | 29,537 (14.8) | 20.8 (20.2, 21.3) | 28,343 (15.5) | 20.8 (20.3, 21.4) | 1194 (7.4) | 18.8 (16.4, 21.3) |
| | Retired | 37,482 (18.8) | 22.5 (22.0, 23.0) | 30,955 (16.9) | 23.5 (23.0, 24.1) | 6527 (40.4) | 17.5 (16.5, 18.5) |
| | Working age not in labor[†] | 5606 (2.8) | 21.0 (19.8, 22.2) | 5311 (2.9) | 21.5 (20.3, 22.8) | 295 (1.8) | 11.5 (8.1, 15.9) |
| Clinical severity* | Asymptomatic | 81,584 (40.9) | 19.9 (19.6, 20.2) | 77,057 (42.1) | 20.3 (19.9, 20.6) | 4527 (28.0) | 14.1 (13.1, 15.3) |
| | Mild/moderate | 7602 (3.8) | 19.9 (18.9, 21.0) | 7216 (3.9) | 20.1 (19.1, 21.2) | 386 (2.4) | 16.6 (12.9, 21.0) |
| | Severe or critical | 32 (0.0) | 15.6 (5.9, 34.3) | 22 (0.0) | 13.6 (3.8, 36.4) | 10 (0.1) | 20.0 (4.0, 64.1) |

CI: confidence interval.

[†] People of working age (≥18 years) unemployed or not in the labor force (disabled); *Clinical severity of first infection.

studies in epidemiology (STROBE) recommendations, and the STROBE checklist is provided in the *Supplementary Appendix* (*Appendix 1—table 4*).

## Results

### Demographic characteristics of the study population

Of the 275,896 individuals with RT-PCR-confirmed SARS-CoV-2 infection during the first Omicron variant wave (from April 1 to May 31, 2022) in Pudong, Shanghai, 1227 individuals died from reasons unrelated to COVID-19 between the first infection and November 2022 and were excluded from our analysis. Most first infections (243,906, 88.8%) occurred in April; for 306 (0.1%) individuals, information on the date of first infection was not available. In April 2022, more than half of the study population (68.6%) had completed full vaccination and one-third (34.4%) had received booster vaccination.

To assess the effect of post-infection vaccination, the analytic sample consisted of 199,312 individuals (*Figure 1*). 85,804 were women (43.1%); 836 (0.4%) had sex information missing. 38.1% of the study participants were aged 20–39 years and only 0.9% were aged 0–6 years (see *Table 1* and *Appendix 1—table 5* for additional information).

### Vaccination after first Omicron variant infection

*Figure 2* (Panel A) shows vaccination coverage in the analytic population over time. In April 2022, a series of emergency epidemic prevention measures, including mass screening with Nucleic Acid Amplification Tests (NAATs), city-wide lockdown, home quarantine of residents, were implemented

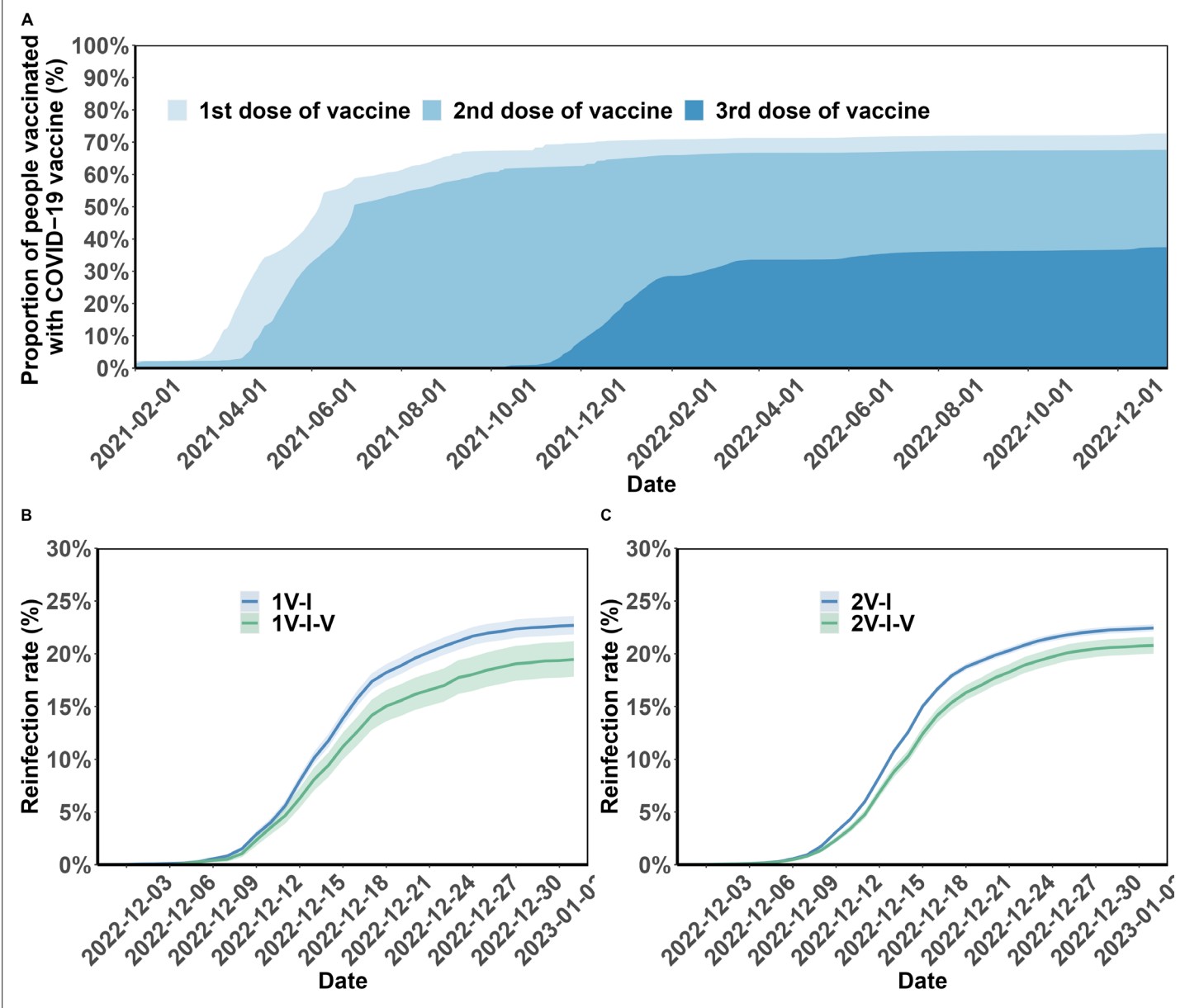

**Figure 2.** Vaccination coverage and cumulative incidence of severe acute respiratory syndrome coronavirus 2 (SARS-CoV-2) reinfection in the study population. Panel A presents the percentages of the study population vaccinated over time. The cumulative incidence of SARS-CoV-2 reinfections is presented by the number of vaccination doses before (panels) and after (lines) first infection (Panels B and C). Shaded regions: 95% confidence intervals (CIs). 1V-I and 2V-I represented 1 and 2 vaccine doses before infection, respectively; 1V-I-V and 2V-I-V correspond to 1 and 2 doses before infection, then post-infection vaccination, respectively. As mentioned in the *Results* section, 142 and 144 study participants who received one and two pre-infection vaccine doses received two post-infection vaccine doses. We do not show the corresponding plot for those individuals who received three pre-infection doses as their post-infection dose was after the start of the follow-up, in December.

in Shanghai, leading to the temporary suspension of vaccination in April and May 2022. By the end of the study period, January 2023, 69.5% (190,815/274,363) of the study participants had completed full vaccination, and 38.4% booster vaccination (*Appendix 1—figure 1*). Vaccination coverages for the first, second, third, and fourth vaccine doses were 72.7%, 67.6%, 37.4%, and 0.3%, respectively.

As mentioned above, only participants who had received vaccination before the first infection were included in this analysis. After infection, 10,241 (5.1%), 5096 (2.6%), and 810 (0.4%) individuals received another vaccine dose in August, and during the periods from September to November 2022 and from December 2022 to January 2023, respectively. Between their first infection, in April–May

2022, and January 2023, 17.4% (2466/14,131) of the study participants who had one pre-infection dose received a second dose of COVID-19 vaccine, and 13.8% (12,886/93,087) of those who had received two pre-infection doses received a third vaccine dose. Only 1.0% (142/14,131) of the study participants who had one pre-infection dose received two post-infection vaccine doses, and 0.2% (144/93,087) of those who had received two pre-infection doses received two post-infection doses (*Appendix 1—table 6*). All individuals with three pre-infection vaccine doses who received a fourth dose (795/92,094, 0.9%) received the post-infection dose in December 2022; for 793 of these participants, the post-infection vaccination received in December was their second booster vaccination (for the other 2, their fourth dose was the first booster vaccination; see *Supplementary Appendix* for more information about vaccine policy in Shanghai).

## SARS-CoV-2 reinfections

Among the study participants, 48,651/199,312 (24.4%) had SARS-CoV-2 reinfection. The median age of individuals with no evidence of reinfection was 41.7 years (IQR: 31.0, 55.7 years), and of individuals with confirmed SARS-CoV-2 reinfection, 45.8 years (IQR: 34.0, 55.9 years) (*Appendix 1—table 5*). The median time interval between the first infection and reinfection was 244.7 days (IQR: 237.4, 250.0), which implies that the risk of misclassifying a long infection as a reinfection was low.

Figure 2 (Panels B and C) shows the cumulative incidence of SARS-CoV-2 reinfection by vaccination status. Overall, individuals who were not vaccinated after their first infection were more often reinfected compared to those who received post-infection vaccination. The percentage of female individuals who became reinfected was lower than that of male individuals (22.4% vs 26.1%; *Table 1*); and reinfection was more common in participants aged 40–59 years compared to those aged 20–39 (29.4% vs 22.4%). Individuals originally from other provinces had a slightly higher risk of reinfection compared to individuals from Shanghai (20.9% vs 18.9%). The risks for retired individuals, for those of working age who were not working, and for those working were 22.5, 21.0, and 20.8%, respectively.

## Vaccine effectiveness

For individuals who had received at least one pre-infection vaccine dose, post-infection vaccination was protective against reinfection (aHR 0.82, 95% confidence interval [CI] 0.79, 0.85). As shown in *Figure 3*, this protective effect was observed in subgroups defined by the number of pre-infection vaccine doses: aHR of 0.84 (95% CI, 0.76, 0.93) and 0.87 (95% CI, 0.83, 0.90) for one and two pre-infection doses, respectively; and for patients with three vaccine doses prior to infection, the association was not statistically significant (aHR 0.96 [0.74, 1.23]). When analyses are stratified by partial and full vaccination status before the first infection, post-infection vaccination was protective (aHR 0.76 [0.68, 0.84] and 0.93 [0.89, 0.97], respectively); and among individuals who had received booster vaccination before the first Omicron variant wave in Shanghai, the hazard ratio estimate was consistent with a more limited effect (aHR 0.95 [0.75, 1.22]). Note that the categories of pre-infection vaccination status and the number of pre-infection vaccinations overlap (*Appendix 1—table 7*). For comparison, results for individuals who had not been vaccinated before their first infection are shown in the *Supplementary Appendix* (supplementary section 'Effect of post-infection vaccination in individuals with no history of vaccination before infection' and *Appendix 1—table 8*).

In analyses stratified by demographic characteristics (*Figure 4*), post-infection vaccine doses were protective in both female (aHR 0.81 [0.76, 0.86]) and male individuals (aHR 0.83 [0.79, 0.87]). Post-infection vaccination was more protective for participants aged 60 years or older (aHR 0.73 [0.69, 0.78]) compared to other age groups. The estimated aHR for individuals who were asymptomatic during their first infection was 0.80 (0.74, 0.87), and for those who were symptomatic during the first infection was 1.01 (0.78, 1.29).

As a secondary analysis, we estimated vaccine effects by calendar time of vaccination. Post-infection vaccine doses given within 30 and 90 days of the second Omicron variant wave were associated with lower hazard of reinfection (aHRs 0.51 [0.44, 0.58] and 0.67 [0.61, 0.74], respectively). Note that in this secondary analysis, exposed and unexposed individuals were matched using propensity scores. We also performed a secondary analysis by vaccine type: for example, we compared patients who completed full vaccination before first infection and received a post-infection Ad5-nCOV vaccine dose, with those who completed full vaccination before first infection and did not receive post-infection vaccination; this approach was repeated for inactivated vaccines and recombinant protein

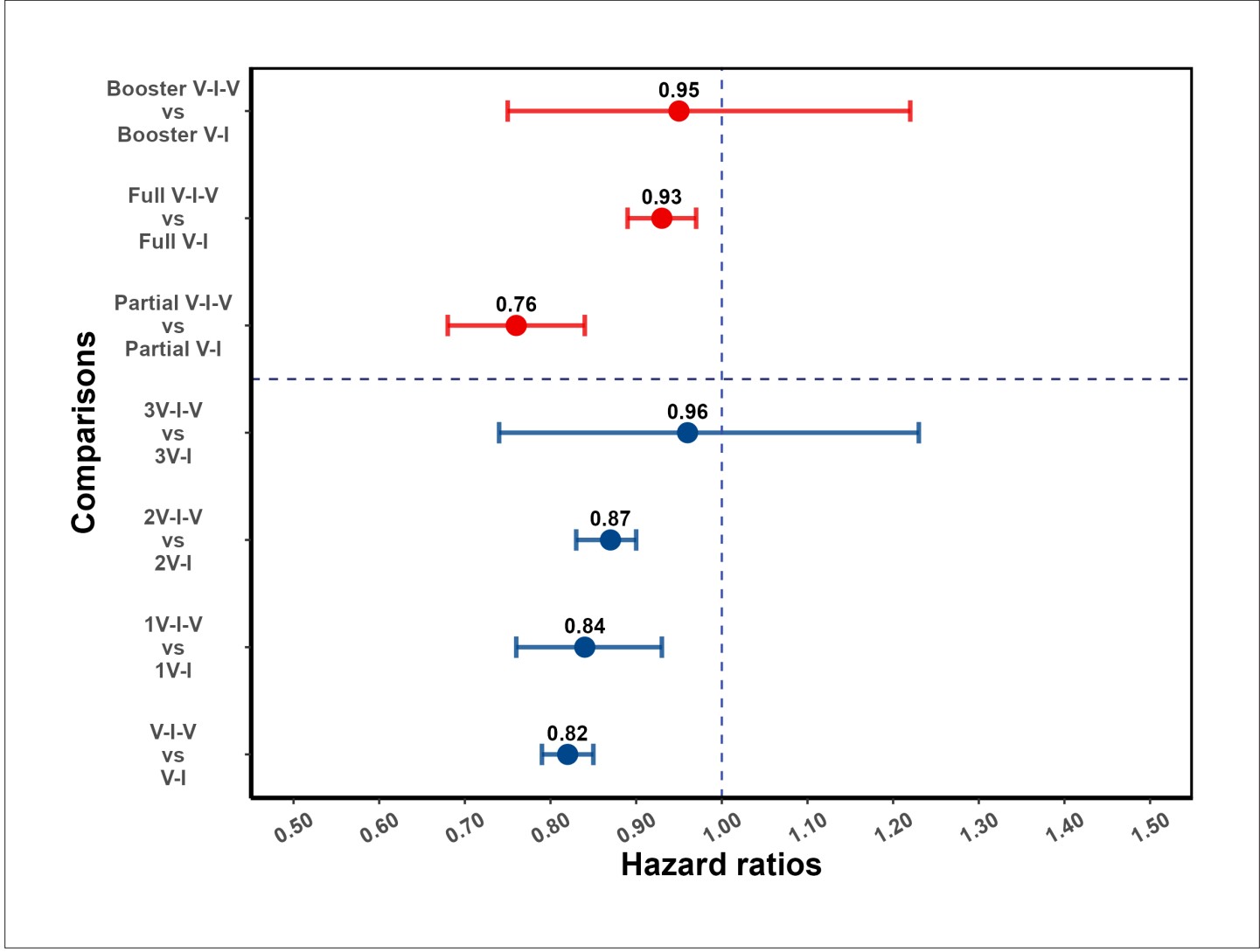

**Figure 3.** Effect of post-infection vaccination on severe acute respiratory syndrome coronavirus 2 (SARS-CoV-2) reinfection stratified by pre-infection vaccination. Error bars (95% confidence intervals [CIs]) and circles represent adjusted hazard ratio (aHR) for SARS-CoV-2 reinfection estimated using Cox proportional hazards models. V-I-V, 1V-I-V, 2V-I-V, and 3V-I-V correspond to any pre-infection vaccination, 1, 2, and 3 vaccine doses before infection, then vaccination, respectively; they were compared to V-I, 1V-I, 2V-I, and 3V-I, respectively. Partial V-I-V, Full V-I-V, and Booster V-I-V represent partial vaccination, full vaccination, and booster vaccination before infection, followed by post-infection vaccination, respectively. The number of doses received by individuals with partial versus full (and full with booster) vaccination depends on the type of SARS-CoV-2 vaccine received; in *Appendix 1—table 3* we present a cross-classification of participants in the analytic population by these vaccination-related categorical variables.

vaccines (*Appendix 1—table 9*). In this analysis, the types of vaccines administered before infection often aligned between the post-infection vaccination group and the post-infection non-vaccination group. Post-infection Ad5-nCoV vaccine dose given to those had received full vaccination before the first infection was associated with lower hazard of reinfection (0.67, 95% CI: 0.56, 0.80); a protective effect was also observed for other vaccine types: inactivated vaccines (0.92, 95% CI: 0.87, 0.98) or recombinant protein vaccines (0.77, 95% CI: 0.64, 0.92). We performed an additional ad hoc sensitivity analysis using a Cox model that was not adjusted for the severity of the first infection. As shown in *Appendix 1—figure 2*, our findings were not affected by the non-inclusion of disease severity in the survival analysis model.

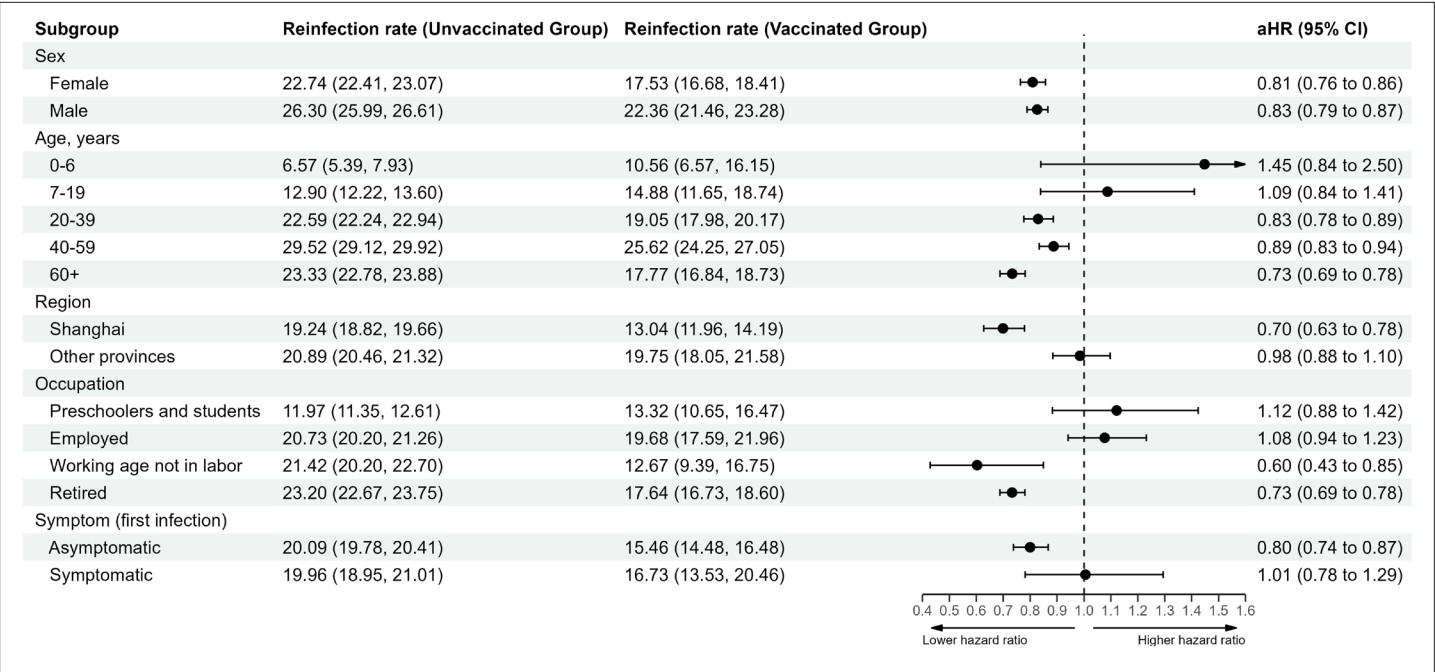

**Figure 4.** Vaccine-related protection against severe acute respiratory syndrome coronavirus 2 (SARS-CoV-2) reinfection stratified by demographic characteristic. The vertical dotted line at 1.0 indicates no effect on protection.

## Discussion

We aimed to estimate the added value of a vaccine dose given after SARS-CoV-2 infection in individuals who had received COVID-19 vaccination before the emergence of the Omicron variant. Although quantification of the protective effect of post-infection vaccination in individuals who have been both exposed to infection and previously vaccinated is a public health policy priority, as in many settings this group represents a large fraction of the population, studies on this question are often complicated by the variable timing of infection, vaccination, and reinfection. Here, we leveraged the epidemiological history of COVID-19 in Shanghai municipality, where exposure to first and second SARS-CoV-2 infections occurred during well-defined periods of time, and large electronic databases allowed analyses of vaccination and infection history. Our study showed that post-infection vaccination reduced the risk of reinfection with the Omicron variant during a second, large surge of infections in the city. We also assessed the impact of the timing of vaccination relative to the second Omicron variant wave, and presented secondary and sensitivity analyses. This study will be used to guide COVID-19 vaccination policy in the municipality of Shanghai, and have implications for the rest of the country.

In Shanghai, COVID-19 vaccination started in February 2021 for individuals aged between 18 and 59 years, and over the following months, both older residents and children were included in the vaccination program (*Huang et al., 2022*). By the time of the first surge of Omicron variant infections, less than 70% of Shanghai residents aged 60–79 years had completed a primary vaccination series, and 40% had received a booster dose (*Huang et al., 2023*). This was the context in which the population in this municipality was exposed to a first large epidemic of SARS-CoV-2 infections. The change in policy in December 2022 that was associated with a significant increase in the number of infections was our primary motivation to study the additional protection afforded by post-infection vaccination. Indeed, although epidemiological studies have been reported on the effect of vaccination against reinfection, primarily in Europe (*Nielsen et al., 2022*), North America (*Bobrovitz et al., 2023*), and Middle East (*Gazit et al., 2023*), there is limited evidence from settings similar to that in Shanghai, where both the infection and vaccination histories, in terms of vaccine types, differ considerably from those in most Western countries. As individuals increasingly question the benefit of getting vaccinated after COVID-19 recovery, evidence generated locally is necessary.

Previous studies have shown that infection alone can result in immune responses that protect against Omicron variant reinfection. For example, a study in Qatar, with a test-negative design,

found that an earlier Omicron variant infection provided protection against symptomatic BA.4 or BA.5 reinfection (*Altarawneh et al., 2022*). There are also different types of evidence that support post-infection vaccination. In Israel, post-infection vaccination protected against reinfection in individuals with no history of pre-infection vaccination (*Hammerman et al., 2022*), and prior SARS-CoV-2 infection was associated with lower risk for breakthrough infection among individuals receiving the BNT162b2 or mRNA-1273 vaccines in Qatar (*Abu-Raddad et al., 2021*). Furthermore, a recent cohort study, performed in the US, of previously infected individuals who were unvaccinated at the time of first infection suggest that vaccination after recovery from COVID-19 decreased risk of reinfection by approximately half (*Lewis et al., 2022*), which was consistent with a case–control study also conducted in the US (*Cavanaugh et al., 2021*). Although our research question was different from those in these studies, data from Shanghai suggest that post-infection vaccination in individuals who had received at least one dose of SARS-CoV-2 vaccine before infection provides protection against reinfection. We note that while post-infection vaccination was protective for individuals who had received one or two vaccine doses before the first Omicron variant wave, for those individuals who had received three pre-infection vaccine doses, the evidence for a protective effect was limited. A possible explanation relates to the timing of post-infection vaccination: for all individuals with three pre-infection vaccine doses, the post-infection dose was given in December 2022 and, as a few weeks are required for immunologic boosting, the relatively short follow-up might have been insufficient to detect an effect. Similarly, evidence that post-infection vaccination was protective for individuals who had received booster vaccination before the spread of the first Omicron variant wave in Shanghai was limited; the majority (793/810) of these individuals had three doses before first Omicron variant infection.

In a secondary analysis that assessed the impact of the timing of the post-infection vaccination on the estimated effect, we observe that vaccination within 30 and 90 days before the second surge of Omicron variant provided different degrees of protection, consistent with waning of immunity even in individuals with multiple vaccine doses and history of infection (*Ferdinands et al., 2022*). As the duration of follow-up in our study was relatively short, we were not able to analyze immunity duration over longer periods of time after the start of the second Omicron variant wave. This observation has implications for COVID-19 vaccination programs: for example, under perfectly functioning logistics, it is possible that timing of vaccination campaigns, including for individuals who have been both previously infected and vaccinated, might be optimal if launched immediately after increase in incidence (e.g. linked to a new variant), rather than immediately after the end of a wave of infections.

In our analyses stratified by demographic characteristics, the protective effect of post-infection vaccination was higher among patients who were 60 years of age or older than among younger individuals. As older individuals are more likely to suffer severe disease, including in settings similar to Shanghai (*Chen et al., 2022b*), this finding suggests that vaccination of individuals in this age group, even after previous infection and vaccination, is beneficial. Similar to the findings of *Sotoodeh Ghorbani et al., 2022*, we also found that the rate of reinfection in females was lower than in males; however, the protective effect of post-infection vaccination was similar in the two groups.

We also observed greater protection after an Ad5-nCoV post-infection vaccination in individuals who were fully vaccinated before their first SARS-CoV-2 infection compared to post-infection vaccination with other vaccine types. Note however that not all post-infection doses were of the same type of pre-infection vaccine doses. Another stratified analysis revealed that the protective effect of additional doses was different in individuals who had asymptomatic presentations of the first infection versus those who had symptomatic disease (aHR 0.80 [0.74, 0.87] and 1.01 [0.78, 1.29]). This could be related to different immune responses after infections with different severities. Indeed, reduced antibody response after the initial SARS-CoV-2 infection has been associated with incidence of reinfections. For example, in the study by *Islamoglu et al., 2021*, it was observed that antibody responses against SARS-CoV-2 were protective against COVID-19 reinfection. Consistently, in the Republic of Korea, CD4[+] T-cell responses tended to be greater in patients who had severe disease (*Kang et al., 2021*). In agreement with these studies, that had different designs, our analysis suggests that individuals who had severe or critical COVID-19 had a lower risk of reinfection compared to those who were asymptomatic (see *Table 1*). Note that individuals who were asymptomatic and those with mild symptoms during the first infection had comparable rates of reinfection. However, individuals with unknown disease severity during the first infection had a higher reinfection rate compared to those with documented clinical severity.

Strengths of our study include: the well-defined timing of first infections and reinfections in Shanghai, linked to the two Omicron variant waves; detailed information on vaccination based on a citywide system; mass screening for SARS-CoV-2 infections during the first Omicron variant surge; free testing during the second Omicron variant wave. Furthermore, to prevent immortal time bias, a common problem in epidemiological studies where treatment/exposure assignment is not aligned with the start of follow-up, we performed survival analysis that used a time varying exposure variable. The insights from our study are also likely generalizable to similar urban settings facing distinct waves of SARS-CoV-2 infections and that adopted the same vaccine types. However, our study also has limitations. First, information on key variables, such as occupation, household registration, clinical severity during the first infection, was missing for a non-negligible fraction of the study population. Although in the primary analysis we performed adjustment for likely confounders, and in the secondary analysis we used propensity score matching, as in other observational studies, we cannot rule out the possibility of residual confounding, which in this context could be related to comorbidities that might affect both the decision to vaccinate after infection and risk of reinfection. Differences in healthcare-seeking behavior could also bias case ascertainment between post-infection vaccinated and unvaccinated individuals (*Yung et al., 2023*). Although we restricted the study population to individuals who had received at least one pre-infection vaccination, which might have led to a higher degree of homogeneity in healthcare-seeking behavior compared to that in the total population, it is possible that this bias might have affected our estimates. For example: individuals who were more health conscious might have been more likely to receive post-infection vaccination and also more likely to seek medical care or testing when reinfected, and this would have biased results toward the null; it is, however, also conceivable that these individuals were more likely to avoid contact with potentially infectious persons, which could have biased results in the opposite direction. Finally, data on the severity of infections during the second wave were not available, which prevented analyses of clinical outcomes other than infections (e.g. COVID-19-related hospitalization or death). Although some previous studies (*Magen et al., 2022*; *Nasreen et al., 2022*; *Sacco et al., 2022*) estimated similar or higher vaccine effectiveness against severe outcomes compared to outcomes that presumably include both milder and severe presentations, this pattern was not observed in all studies. Epidemiologists and public health officials who will use our results to define vaccination policy should thus take into account the fact that our analysis does not capture all benefits of post-infection vaccinations.

## Conclusions

Although SARS-CoV-2 vaccination, including booster doses, is recommended by the public health authorities in Shanghai to reduce the local disease burden caused by COVID-19, there is increasing unwillingness in the population to receive additional vaccine doses, and vaccine fatigue has been frequently reported. Our study provides evidence that there is additional value for individuals who have been vaccinated in receiving vaccine doses after infections. It also suggests that vaccination programs need to be linked to efficient surveillance for new infections so that public health authorities can maximize impact of additional doses, including in this group of patients.

## Acknowledgements

We gratefully acknowledge all participants in this study.

## Additional information

### Competing interests

Pengfei Deng: Jie Tian: The other authors declare that no competing interests exist.

### Funding

| Funder | Grant reference number | Author |
| --- | --- | --- |
| Key Discipline Program of Pudong New Area Health System | PWZxk2022-25 | Pengfei Deng |

| Funder | Grant reference number | Author |
| --- | --- | --- |
| Development and Application of Intelligent Epidemic Surveillance and Artificial Intelligence Analysis System | 21002411400 | Weibing Wang |
| Shanghai Three-year Action Plan to Strengthen the Construction of Public Health System | GWVI-11.2-XD08 | Caoyi Xue |
| Shanghai Municipal Health Commission | GWVI-11.1-02 | Caoyi Xue |
| Shanghai Municipal Health Commission | 20214Y0020 | Ye Yao |
| Natural Science Foundation of Shanghai Municipality | 22ZR1414600 | Ye Yao |
| Young Health Talents Program of Shanghai Municipality | 2022YQ076 | Ye Yao |

The funders had no role in study design, data collection and interpretation, or the decision to submit the work for publication.

## Author contributions

Bo Zheng, Conceptualization, Formal analysis, Methodology, Visualization, Writing – original draft; Bronner P Gonçalves, Conceptualization, Formal analysis, Methodology, Writing – original draft, Writing – review and editing; Pengfei Deng, Data curation, Formal analysis, Funding acquisition, Investigation, Methodology, Project administration, Writing – original draft; Weibing Wang, Funding acquisition, Resources, Supervision, Writing – review and editing; Jie Tian, Conceptualization, Software, Visualization; Xueyao Liang, Conceptualization, Data curation; Ye Yao, Data curation, Resources, Validation, Writing – review and editing; Caoyi Xue, Data curation, Funding acquisition, Resources, Validation, Writing – review and editing

## Author ORCIDs

Bronner P Gonçalves ⬤ https://orcid.org/0000-0002-3329-6050
Weibing Wang ⬤ https://orcid.org/0000-0002-4497-5251
Ye Yao ⬤ https://orcid.org/0000-0002-0878-8942

## Ethics

This study has been approved by the Health Committee of Pudong New Area and the Ethics Review Committee of Shanghai Pudong New Area Center for Disease Control and Prevention (approval number: PDCDCLL-20220801-001). The Ethics Committee believes that the study strictly and fully embodies the rights of participants, respects their right to know and privacy, guarantees the safety and welfare of participants, and conforms to the current policies and regulations of biomedical ethical research involving human beings in China. The Shanghai Pudong New Area Health Commission has the right to collect and use epidemiological data of COVID-19 patients as part of the investigation of infectious disease outbreaks. A completely anonymous design was adopted in this study, which was approved by Shanghai Pudong New Area Health Committee. Therefore, the individual consent of the patient is not required.

Reviewer #1 (Public review): https://doi.org/10.7554/eLife.94990.3.sa1
Reviewer #2 (Public review): https://doi.org/10.7554/eLife.94990.3.sa2
Author response https://doi.org/10.7554/eLife.94990.3.sa3

## Additional files

### Supplementary files

MDAR checklist

1121Source code 1. Data analysis and graph plotting for Cox models.

### Data availability

Due to restrictions on data that may potentially reveal identifiable or sensitive patient information, and in accordance with a confidentiality agreement with the Pudong New Area Center for Disease Control and Prevention in Shanghai, the original datasets used and/or analyzed in this study are not publicly available. However, the data that support the findings of this study can be requested for non-commercial purposes by interested researchers. Access to the data is subject to approval by the Ethical Review Committee of the Pudong New Area Center for Disease Control and Prevention in Shanghai, which will then provide the data directly to researchers who fulfill the criteria for access to confidential data. The application process does not require the submission of a project proposal but does require a detailed description of the researcher's identity, the intended use of the data, the purpose of the study, and a signed data confidentiality agreement to safeguard the personal information of patients. Requests for data access should be directed to the Ethical Committee at masterdpf@163.com, who serve as the data custodians. The code utilized for data analysis and graph plotting is included in *Source code 1*. The anonymized case reinfection data can be accessed at https://doi.org/10.5061/dryad.rfj6q57ks.

The following dataset was generated:

| Author(s) | Year | Dataset title | Dataset URL | Database and Identifier |
|---|---|---|---|---|
| Yao Y, Zheng B | 2024 | COVID-19 reinfection data on individuals diagnosed with their first SARS-CoV-2 infection | https://doi.org/10.5061/dryad.rfj6q57ks | Dryad Digital Repository, 10.5061/dryad.rfj6q57ks |

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

## Appendix 1

## Introduction of COVID-19 vaccination procedures

Vaccination status was categorized into four levels in accordance with the national technical recommendations for COVID-19 vaccination: (1) unvaccinated—that is, no history of COVID-19 vaccination before first infection date; (2) partial vaccination—either one dose of inactivated vaccine, or two doses of recombinant protein vaccine (three doses are recommended for primary vaccination) before the first infection; (3) full primary vaccination—either two doses of inactivated vaccine, one dose of adenovirus vector vaccine, three doses of recombinant protein vaccine before first infection; or (4) booster vaccination—either two doses of Ad5-vectored vaccine; or two doses of inactivated vaccine and one booster dose of inactivated vaccine, or adenovirus vector vaccine; or three doses of recombinant protein and one booster dose of recombinant protein, or adenovirus vector vaccine before first infection date.

During the second Omicron wave in December 2022, Shanghai launched the implementation plan of the second dose of enhanced immunization of the novel coronavirus vaccine, encouraging second dose of enhanced immunization in high-risk groups: elderly people over 60 years old, people with serious underlying diseases and people with low immunity on the basis of the first dose of enhanced immunization.

**Appendix 1—table 1.** Baseline demographic characteristics of unvaccinated and vaccinated groups after first infection stratified by dose received before first infection.
This comparison only refers to the secondary analyses where exposed and unexposed individuals were matched based on propensity scores.

| Characteristics | | Received 1 vaccine dose prior first infection | | | Received 2 vaccine doses prior first infection | | |
|---|---|---|---|---|---|---|---|
| | | Unvaccinated group (N = 2247) | Vaccinated group (N = 2247) | SMD | Unvaccinated group (N = 12,674) | Vaccinated group (N = 12,674) | SMD |
| Sex | Male | 1309 (58.3) | 1309 (58.3) | <0.001 | 6784 (53.5) | 6784 (53.5) | <0.001 |
| | Female | 924 (41.1) | 924 (41.1) | | 5841 (46.1) | 5841 (46.1) | |
| | Unknown | 14 (0.6) | 14 (0.6) | | 49 (0.4) | 49 (0.4) | |
| Age, years | 0–6 | 78 (3.5) | 78 (3.5) | <0.001 | 89 (0.7) | 89 (0.7) | <0.001 |
| | 7–19 | 155 (6.9) | 155 (6.9) | | 258 (2.0) | 258 (2.0) | |
| | 20–39 | 819 (36.4) | 819 (36.4) | | 3828 (30.2) | 3828 (30.2) | |
| | 40–59 | 479 (21.3) | 479 (21.3) | | 3323 (26.2) | 3323 (26.2) | |
| | 60+ | 709 (31.6) | 709 (31.6) | | 5142 (40.6) | 5142 (40.6) | |
| | Unknown | 7 (0.3) | 7 (0.3) | | 34 (0.3) | 34 (0.3) | |
| Regions | Shanghai | 370 (16.5) | 370 (16.5) | <0.001 | 2369 (18.7) | 2369 (18.7) | <0.001 |
| | Other provinces | 296 (13.2) | 296 (13.2) | | 1481 (11.7) | 1481 (11.7) | |
| | Unknown | 1581 (70.4) | 1581 (70.4) | | 8824 (69.6) | 8824 (69.6) | |
| Occupations | Preschoolers and students | 228 (10.1) | 228 (10.1) | <0.001 | 326 (2.6) | 326 (2.6) | <0.001 |
| | Employed | 161 (7.2) | 161 (7.2) | | 865 (6.8) | 865 (6.8) | |
| | Working age not in labor | 31 (1.4) | 31 (1.4) | | 245 (1.9) | 245 (1.9) | |
| | Retired | 716 (31.9) | 716 (31.9) | | 5205 (41.1) | 5205 (41.1) | |
| | Unknown | 1111 (49.4) | 1111 (49.4) | | 6033 (47.6) | 6033 (47.6) | |

*Appendix 1—table 1 Continued on next page*

*Appendix 1—table 1 Continued*

| Characteristics | | Received 1 vaccine dose prior first infection | | | Received 2 vaccine doses prior first infection | | |
|---|---|---|---|---|---|---|---|
| | | Unvaccinated group (N = 2247) | Vaccinated group (N = 2247) | SMD | Unvaccinated group (N = 12,674) | Vaccinated group (N = 12,674) | SMD |
| Clinical severity | Asymptomatic | 612 (27.2) | 612 (27.2) | 0.043 | 3546 (28.0) | 3546 (28.0) | 0.016 |
| | Mild and moderate | 54 (2.4) | 52 (2.3) | | 301 (2.4) | 297 (2.3) | |
| | Severe or critical | 0 (0.0) | 2 (0.1) | | 3 (0.0) | 7 (0.1) | |
| | Unknown | 1581 (70.4) | 1581 (70.4) | | 8824 (69.6) | 8824 (69.6) | |

SMD: standardized mean difference.

**Appendix 1—table 2.** Baseline demographic characteristics of unvaccinated and vaccinated groups after first infection stratified by intervals before the policy change.
This comparison only refers to the secondary analyses where exposed and unexposed individuals were matched based on propensity scores.

| Characteristics | | Within 30 days* | | | Within 90 days* | | |
|---|---|---|---|---|---|---|---|
| | | Unvaccinated group (N = 3137) | Vaccinated group (N = 3137) | SMD | Unvaccinated group (N = 4161) | Vaccinated group (N = 4161) | SMD |
| Sex | Male | 1585 (50.5) | 1585 (50.5) | <0.001 | 2275 (54.7) | 2275 (54.7) | <0.001 |
| | Female | 1541 (49.1) | 1541 (49.1) | | 1865 (44.8) | 1865 (44.8) | |
| | Unknown | 11 (0.4) | 11 (0.4) | | 21 (0.5) | 21 (0.5) | |
| Age, years | 0–6 | 69 (2.2) | 69 (2.2) | <0.001 | 77 (1.9) | 77 (1.9) | <0.001 |
| | 7–19 | 56 (1.8) | 56 (1.8) | | 107 (2.6) | 107 (2.6) | |
| | 20–39 | 870 (27.7) | 870 (27.7) | | 1472 (35.4) | 1472 (35.4) | |
| | 40–59 | 645 (20.6) | 645 (20.6) | | 1003 (24.1) | 1003 (24.1) | |
| | 60+ | 1495 (47.7) | 1495 (47.7) | | 1494 (35.9) | 1494 (35.9) | |
| | Unknown | 2 (0.1) | 2 (0.1) | | 8 (0.2) | 8 (0.2) | |
| Regions | Shanghai | 1015 (32.4) | 1015 (32.4) | <0.001 | 1060 (25.5) | 1060 (25.5) | <0.001 |
| | Other provinces | 341 (10.9) | 341 (10.9) | | 646 (15.5) | 646 (15.5) | |
| | Unknown | 1781 (56.8) | 1781 (56.8) | | 2455 (59.0) | 2455 (59.0) | |
| Occupations | Preschoolers and students | 124 (4.0) | 124 (4.0) | <0.001 | 180 (4.3) | 180 (4.3) | <0.001 |
| | Employed | 181 (5.8) | 181 (5.8) | | 344 (8.3) | 344 (8.3) | |
| | Working age not in labor | 54 (1.7) | 54 (1.7) | | 81 (1.9) | 81 (1.9) | |
| | Retired | 1526 (48.6) | 1526 (48.6) | | 1528 (36.7) | 1528 (36.7) | |
| | Unknown | 1252 (39.9) | 1252 (39.9) | | 2028 (48.7) | 2028 (48.7) | |

*Appendix 1—table 2 Continued on next page*

*Appendix 1—table 2 Continued*

| Characteristics | | Within 30 days* | | | Within 90 days* | | |
|---|---|---|---|---|---|---|---|
| | | Unvaccinated group (N = 3137) | Vaccinated group (N = 3137) | SMD | Unvaccinated group (N = 4161) | Vaccinated group (N = 4161) | SMD |
| Clinical severity | Asymptomatic | 1273 (40.6) | 1273 (40.6) | 0.01 | 1600 (38.5) | 1600 (38.5) | <0.001 |
| | Mild and moderate | 79 (2.5) | 80 (2.6) | | 102 (2.5) | 102 (2.5) | |
| | Severe or critical | 4 (0.1) | 3 (0.1) | | 4 (0.1) | 4 (0.1) | |
| | Unknown | 1781 (56.8) | 1781 (56.8) | | 2455 (59.0) | 2455 (59.0) | |

SMD: standardized mean difference. *Before second Omicron variant wave

**Appendix 1—table 3.** Baseline demographic characteristics of unvaccinated and vaccinated groups after first infection stratified by vaccination status prior infection.

This comparison only refers to the secondary analyses where exposed and unexposed individuals were matched based on propensity scores.

| Characteristics | | Partial vaccination prior to infection | | | Full vaccination prior to infection | | |
|---|---|---|---|---|---|---|---|
| | | Unvaccinated group (N = 2451) | Vaccinated group (N = 2451) | SMD | Unvaccinated group (N = 11,034) | Vaccinated group (N = 11,034) | SMD |
| Sex | Male | 1286 (52.5) | 1286 (52.5) | <0.001 | 6133 (55.6) | 6133 (55.6) | <0.001 |
| | Female | 1148 (46.8) | 1148 (46.8) | | 4859 (44.0) | 4859 (44.0) | |
| | Unknown | 17 (0.7) | 17 (0.7) | | 42 (0.4) | 42 (0.4) | |
| Age, years | 0–6 | 163 (6.7) | 163 (6.7) | <0.001 | 181 (1.6) | 181 (1.6) | <0.001 |
| | 7–19 | 232 (9.5) | 232 (9.5) | | 3722 (33.7) | 3722 (33.7) | |
| | 20–39 | 905 (36.9) | 905 (36.9) | | 3186 (28.9) | 3186 (28.9) | |
| | 40–59 | 559 (22.8) | 559 (22.8) | | 3914 (35.5) | 3914 (35.5) | |
| | 60+ | 585 (23.9) | 585 (23.9) | | 31 (0.3) | 31 (0.3) | |
| | Unknown | 7 (0.3) | 7 (0.3) | | 1674 (15.2) | 1674 (15.2) | <0.001 |
| Regions | Shanghai | 460 (18.8) | 460 (18.8) | <0.001 | 1428 (12.9) | 1428 (12.9) | |
| | Other provinces | 330 (13.5) | 330 (13.5) | | 7932 (71.9) | 7932 (71.9) | |
| | Unknown | 1661 (67.8) | 1661 (67.8) | | 168 (1.5) | 168 (1.5) | <0.001 |
| Occupations | Preschoolers and students | 382 (15.6) | 382 (15.6) | <0.001 | 834 (7.6) | 834 (7.6) | |
| | Employed | 184 (7.5) | 184 (7.5) | | 186 (1.7) | 186 (1.7) | |
| | Working age not in labor | 62 (2.5) | 62 (2.5) | | 3959 (35.9) | 3959 (35.9) | |
| | Retired | 593 (24.2) | 593 (24.2) | | 5887 (53.4) | 5887 (53.4) | |
| | Unknown | 1230 (50.2) | 1230 (50.2) | | 2827 (25.6) | 2827 (25.6) | <0.001 |
| Clinical severity | Asymptomatic | 726 (29.6) | 726 (29.6) | 0.039 | 272 (2.5) | 272 (2.5) | |
| | Mild and moderate | 63 (2.6) | 60 (2.4) | | 3 (0.0) | 3 (0.0) | |
| | Severe or critical | 1 (0.0) | 4 (0.2) | | 7932 (71.9) | 7932 (71.9) | |
| | Unknown | 1661 (67.8) | 1661 (67.8) | | 7943 (71.9) | 7943 (71.9) | |

SMD: standardized mean difference.

**Appendix 1—table 4.** STROBE statement—checklist of items that should be included in reports of *cohort studies.*

|  | Item No | Recommendation | Page/Figure |
|---|---|---|---|
| Title and abstract |  | Indicate the study's design with a commonly used term in the title or the abstract | Page 1 |
|  | 1 | Provide in the abstract an informative and balanced summary of what was done and what was found | Page 1 and 2 |
| Introduction |  |  |  |
| Background/rationale | 2 | Explain the scientific background and rationale for the investigation being reported | Page 2 |
| Objectives | 3 | State specific objectives, including any prespecified hypotheses | Page 2 |
| Methods |  |  |  |
| Study design | 4 | Present key elements of study design early in the paper | Page 2 and 3 |
| Setting | 5 | Describe the setting, locations, and relevant dates, including periods of recruitment, exposure, follow-up, and data collection | Page 2 and 3 |
| Participants |  | Give the eligibility criteria, and the sources and methods of selection of participants. Describe methods of follow-up | Page 2 and 3 |
|  | 6 | For matched studies, give matching criteria and number of exposed and unexposed | Page 3 and 4 |
| Variables | 7 | Clearly define all outcomes, exposures, predictors, potential confounders, and effect modifiers. Give diagnostic criteria, if applicable | Page 3 and 4 |
| Data sources/ measurement | 8* | For each variable of interest, give sources of data and details of methods of assessment (measurement). Describe comparability of assessment methods if there is more than one group | Page 3 and 4 |
| Bias | 9 | Describe any efforts to address potential sources of bias | Page 3 and 4 |
| Study size | 10 | Explain how the study size was arrived at | - |
| Quantitative variables | 11 | Explain how quantitative variables were handled in the analyses. If applicable, describe which groupings were chosen and why | Page 3 and 4 |
| Statistical methods |  | Describe all statistical methods, including those used to control for confounding | Page 3 and 4 |
|  |  | Describe any methods used to examine subgroups and interactions | Page 3 and 4 |
|  |  | Explain how missing data were addressed | Page 3 and 4 |
|  |  | If applicable, explain how loss to follow-up was addressed | - |
|  | 12 | Describe any sensitivity analyses | Page 3 and 4 |
| Results |  |  |  |

*Appendix 1—table 4 Continued on next page*

Appendix 1—table 4 Continued

| | Item No | Recommendation | Page/Figure |
|---|---|---|---|
| Participants | | Report numbers of individuals at each stage of study—e.g. numbers potentially eligible, examined for eligibility, confirmed eligible, included in the study, completing follow-up, and analyzed | Page 4 and 5 |
| | | Give reasons for non-participation at each stage | Page 5 |
| | 13* | Consider use of a flow diagram | *Figure 1* |
| Descriptive data | | Give characteristics of study participants (e.g. demographic, clinical, social) and information on exposures and potential confounders | Page 5 |
| | | Indicate number of participants with missing data for each variable of interest | Page 5 |
| | 14* | Summarize follow-up time (e.g. average and total amount) | Page 5 |
| Outcome data | 15* | Report numbers of outcome events or summary measures over time | *Table 1* and *Appendix 1—table 5* |
| Main results | | Give unadjusted estimates and, if applicable, confounder-adjusted estimates and their precision (e.g. 95% confidence interval). Make clear which confounders were adjusted for and why they were included | Page 5 and 6 |
| | | Report category boundaries when continuous variables were categorized | *Figures 2–4* |
| | 16 | If relevant, consider translating estimates of relative risk into absolute risk for a meaningful time period | Page 7 and 8 |
| Other analyses | 17 | Report other analyses done—e.g. analyses of subgroups and interactions, and sensitivity analyses | Page 8 |
| Discussion | | | |
| Key results | 18 | Summarize key results with reference to study objectives | Page 5–8 |
| Limitations | 19 | Discuss limitations of the study, taking into account sources of potential bias or imprecision. Discuss both direction and magnitude of any potential bias | Page 9–11 |
| Interpretation | 20 | Give a cautious overall interpretation of results considering objectives, limitations, multiplicity of analyses, results from similar studies, and other relevant evidence | Page11 |
| Generalisability | 21 | Discuss the generalisability (external validity) of the study results | Page10 |
| Other information | | | |
| Funding | 22 | Give the source of funding and the role of the funders for the present study and, if applicable, for the original study on which the present article is based | Page 11 and 12 |

*Give information separately for exposed and unexposed groups.

**Appendix 1—table 5.** Descriptive overview of the study population.
While in *Table 1* we present information by post-infection vaccination status, in this table data are

shown based on reinfection during the second Omicron variant wave. Note that this table reflects the vaccination status by the end of the study period, January 2023. According to vaccination procedures, boosters are given post-full vaccination, indicating that those with boosters have completed the initial series.

| Characteristic | | N = 199,312 (%) | Not reinfection (n = 150,661), n (%) | Reinfection (n = 48,651), n (%) |
|---|---|---|---|---|
| Median age (IQR), years | | 42.79 (31.77, 55.73) | 41.66 (31.03, 55.70) | 45.82 (33.97, 55.85) |
| Mean age (95% CI), years | | 43.88 (43.8, 43.95) | 43.37 (43.28, 43.45) | 45.45 (45.32, 45.58) |
| Age group, years | 0–6 | 1736 (0.9) | 1615 (1.1) | 121 (0.2) |
| | 7–19 | 10,762 (5.4) | 9363 (6.2) | 1399 (2.9) |
| | 20–39 | 75,955 (38.1) | 58,911 (39.1) | 17,044 (35.0) |
| | 40–59 | 74,680 (37.5) | 52,726 (35.0) | 21,954 (45.1) |
| | ≥ 60 | 35,903 (18.0) | 27,796 (18.4) | 8107 (16.7) |
| Sex | Male | 112,672 (56.5) | 83,263 (55.3) | 29,409 (60.4) |
| | Female | 85,804 (43.1) | 66,581 (44.2) | 19,223 (39.5) |
| Vaccination status | Partial vaccination | 8497 (4.3) | 6968 (4.6) | 1529 (3.1) |
| | Full vaccination | 85,546 (42.9) | 66,486 (44.2) | 19,060 (39.2) |
| | Booster vaccination | 105,269 (52.8) | 77,207 (51.2) | 28,062 (57.7) |
| Clinical severity during the first infection | Asymptomatic | 81,584 (40.9) | 65,333 (43.4) | 16,251 (33.4) |
| | Mild and moderate | 7602 (3.8) | 6087 (4.0) | 1515 (3.1) |
| | Severe or critical | 32 (0.0) | 27 (0.0) | 5 (0.0) |

IQR: interquartile range. Data are presented as means with 95% confidence intervals (95% CIs) or as proportions (%). Note that for the variables on age, sex, and clinical severity, data are missing for fractions of the study population, especially the clinical severity during the first infection.

**Appendix 1—table 6.** Characteristics of vaccination distribution among study population.

| Number of vaccine doses before infection | Participants in the analysis | Participants received post-infection dose | 1 dose after infection | | 2 doses after infection | | |
|---|---|---|---|---|---|---|---|
| | Total | Total | Total | Before Dec. | Total | Dose 1 before Dec. | Dose 2 before Dec. |
| 1 | 14,131 | 2466 | 2324 | 2324 | 142 | 142 | 13 |
| 2 | 93,087 | 12,886 | 12,742 | 12,727 | 144 | 144 | 32 |
| 3 | 92,094 | 795 | 795 | 0 | 0 | 0 | 0 |

**Appendix 1—table 7.** Cross-classification based on the number of vaccine doses received before the first infection and vaccination status.

| Number of vaccine doses before infection | Participants in the analysis | Partial vaccination | Full vaccination | Booster vaccination |
|---|---|---|---|---|
| 1 | 14,131 | 8665 | 5466 | 0 |
| 2 | 93,087 | 2055 | 88,673 | 2359 |
| 3 | 92,094 | 0 | 322 | 91,772 |

**Appendix 1—table 8.** Effect of post-infection vaccination in individuals with no history of vaccination before infection.

This table only include the 74,962 unvaccinated from the analysis and is stratified by demographic characteristic.

| | Subgroup | Reinfection rate (unvaccinated group) | Reinfection rate (vaccinated group) | aHR (95% CI) | p value |
|---|---|---|---|---|---|
| Overall | | 10.88 (10.53, 11.24) | 7.62 (6.47, 8.92) | 1.06 (0.97, 1.16) | 0.216 |
| Sex | Female | 12.63 (12.24, 13.04) | 10.95 (9.75, 12.25) | 0.97 (0.84, 1.11) | 0.653 |
| | Male | 14.10 (13.75, 14.46) | 14.87 (13.41, 16.44) | 1.12 (0.99, 1.26) | 0.070 |
| Age, years | 0–6 | 5.31 (4.69, 5.99) | 9.45 (5.88, 14.46) | 1.89 (1.08, 3.30) | 0.026 |
| | 7–19 | 9.81 (8.86, 10.83) | 11.89 (7.66, 17.68) | 1.32 (0.83, 2.10) | 0.238 |
| | 20–39 | 13.17 (12.72, 13.64) | 16.4 (13.88, 19.25) | 1.12 (0.92, 1.37) | 0.250 |
| | 40–59 | 14.34 (13.82, 14.87) | 14.17 (11.66, 17.07) | 1.34 (1.08, 1.67) | 0.008 |
| | 60+ | 15.04 (14.54, 15.56) | 11.72 (10.57, 12.95) | 0.92 (0.82, 1.04) | 0.204 |
| Region | Shanghai | 11.79 (11.32, 12.27) | 6.74 (5.61, 8.03) | 0.77 (0.61, 0.98) | 0.034 |
| | Other provinces | 9.83 (9.36, 10.32) | 13.47 (9.76, 18.15) | 1.72 (1.21, 2.43) | 0.002 |
| Occupation | Preschoolers and students | 7.21 (6.65, 7.80) | 10.40 (7.51, 14.06) | 1.47 (1.02, 2.12) | 0.041 |
| | Employed | 9.76 (9.17, 10.38) | 17.61 (11.67, 25.57) | 1.84 (1.17, 2.91) | 0.009 |
| | Working age not in labor | 12.55 (11.06, 14.18) | 1.28 (0.26, 4.11) | 0.27 (0.07, 1.10) | 0.069 |
| | Retired | 15.13 (14.63, 15.65) | 11.37 (10.26, 12.57) | 0.90 (0.79, 1.02) | 0.095 |
| Symptom (first infection) | Asymptomatic | 10.88 (10.53, 11.24) | 7.62 (6.47, 8.92) | 0.94 (0.76, 1.15) | 0.551 |
| | Symptomatic | 10.99 (9.87, 12.22) | 9.26 (4.75, 16.43) | 1.12 (0.55, 2.27) | 0.748 |

**Appendix 1—table 9.** Effect of vaccination after first infection against severe acute respiratory syndrome coronavirus 2 (SARS-CoV-2) reinfection stratified by sequence of vaccination dose and vaccination status before first infection.

| Characteristics[#] | | Unvaccinated group | | Vaccinated group | | Number with same type of vaccine before infection, N (%) | aHR (95% CI) |
|---|---|---|---|---|---|---|---|
| | | N (%) | Reinfection rate, % (95% CI) | N (%) | Reinfection rate, % (95% CI) | | |
| 1 dose of vaccine before first infection | Overall | 2247 (100) | 23.63 (21.69, 25.71) | 2247 (100) | 19.94 (18.16, 21.85) | 1284 (57.14) | 0.81 (0.72, 0.92) |
| | Ad5-nCoV | 423 (18.83) | 27.90 (23.20, 33.28) | 423 (18.83) | 26.00 (21.48, 31.21) | 213 (50.35) | 0.91 (0.70, 1.18) |
| | Inactivated vaccine | 1679 (74.72) | 22.75 (20.56, 25.12) | 1679 (74.72) | 18.23 (16.27, 20.35) | 1064 (63.37) | 0.77 (0.66, 0.89) |
| | Recombinant protein vaccine | 12 (0.53) | 33.33 (11.14, 79.25) | 12 (0.53) | 41.67 (15.80, 91.33) | 7 (58.33) | 1.28 (0.34, 4.80) |

*Appendix 1—table 9 Continued on next page*

*Appendix 1—table 9 Continued*

| Characteristics[#] | | Unvaccinated group | | Vaccinated group | | Number with same type of vaccine before infection, N (%) | aHR (95% CI) |
|---|---|---|---|---|---|---|---|
| | | N (%) | Reinfection rate, % (95% CI) | N (%) | Reinfection rate, % (95% CI) | | |
| 2 doses of vaccine before first infection | Overall | 12,674 (100) | 24.57 (23.72, 25.44) | 12,674 (100) | 20.81 (20.03, 21.62) | 11,979 (94.52) | 0.83 (0.79, 0.87) |
| | Ad5-nCoV | 787 (6.21) | 24.27 (21.01, 27.90) | 787 (6.21) | 11.61 (9.40, 14.20) | 512 (65.06) | 0.45 (0.35, 0.58) |
| | Inactivated vaccine | 10,747 (84.80) | 24.60 (23.68, 25.55) | 10,747 (84.80) | 21.39 (20.53, 22.27) | 10,465 (97.38) | 0.85 (0.80, 0.90) |
| | Recombinant protein vaccine | 1057 (8.34) | 25.26 (22.37, 28.43) | 1057 (8.34) | 20.91 (18.29, 23.80) | 1002 (94.80) | 0.79 (0.66, 0.95) |
| Partial vaccination before first infection[†] | Overall | 2451 (100) | 20.52 (18.79, 22.38) | 2451 (100) | 16.40 (14.86, 18.06) | 2349 (95.84) | 0.78 (0.69, 0.89) |
| | Inactivated vaccine | 2343 (95.59) | 20.44 (18.67, 22.34) | 2343 (95.59) | 16.30 (14.73, 18.00) | 2304 (98.34) | 0.78 (0.68, 0.90) |
| | Recombinant protein vaccine | 47 (1.92) | 19.15 (9.46, 34.95) | 47 (1.92) | 17.02 (8.03, 32.12) | 45 (95.74) | 0.92 (0.35, 2.38) |
| Full vaccination before first infection | Overall | 11,034 (100) | 25.33 (24.40, 26.28) | 11,034 (100) | 22.72 (21.84, 23.62) | 9542 (86.48) | 0.88 (0.83, 0.93) |
| | Ad5-nCoV | 1209 (10.96) | 24.32 (21.66, 27.22) | 1209 (10.96) | 17.12 (14.91, 19.58) | 805 (66.58) | 0.67 (0.56, 0.80) |
| | Inactivated vaccine | 8787 (79.64) | 25.37 (24.33, 26.44) | 8787 (79.64) | 23.69 (22.69, 24.73) | 7944 (90.41) | 0.92 (0.87, 0.98) |
| | Recombinant protein vaccine | 1037 (9.40) | 26.13 (23.16, 29.39) | 1037 (9.40) | 20.93 (18.28, 23.85) | 793 (76.47) | 0.77 (0.64, 0.92) |

Data are presented as average (95% CI), or n (n/N%), where N is the total number of patients. [#]Individuals received two doses of inactivated vaccine, recombinant protein vaccine or Ad5-nCoV were excluded. [†]Only few post-infection Ad5-nCoV vaccine dose given to those had partial vaccination before the first infection.

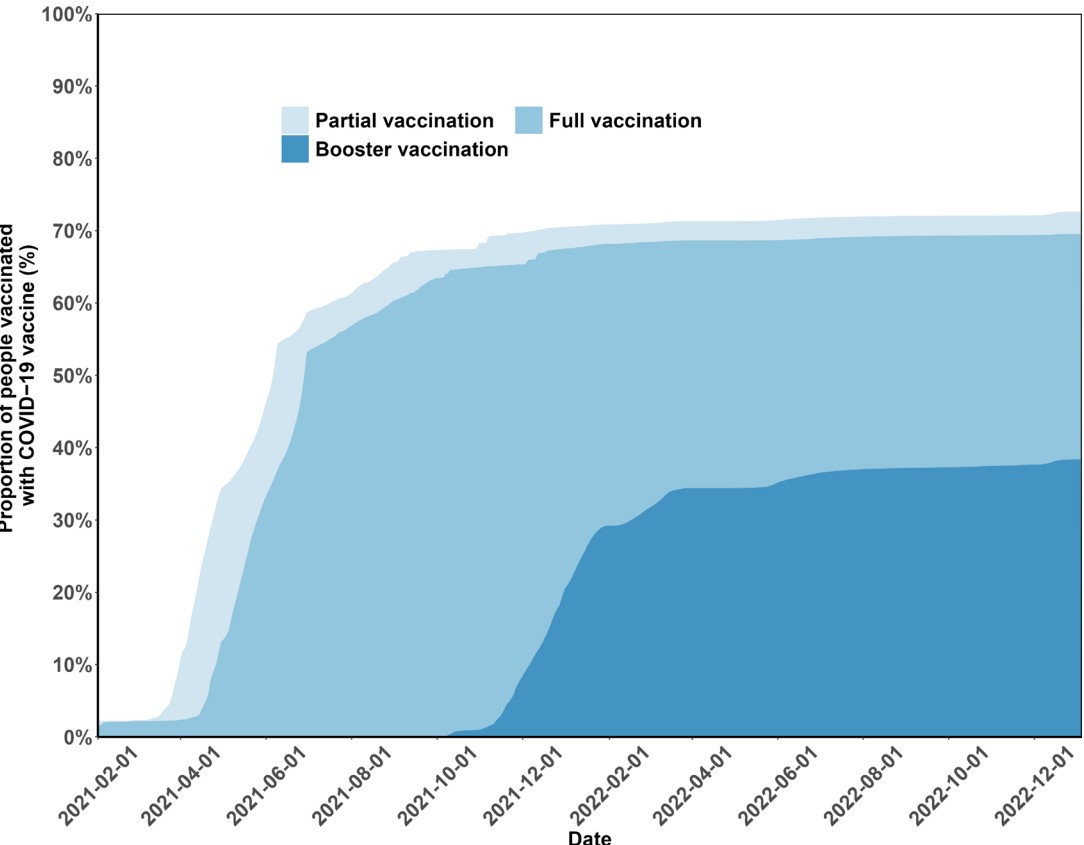

**Appendix 1—figure 1.** Vaccination status coverage in the study population. The figure presents the percentages of the study population vaccinated over time.

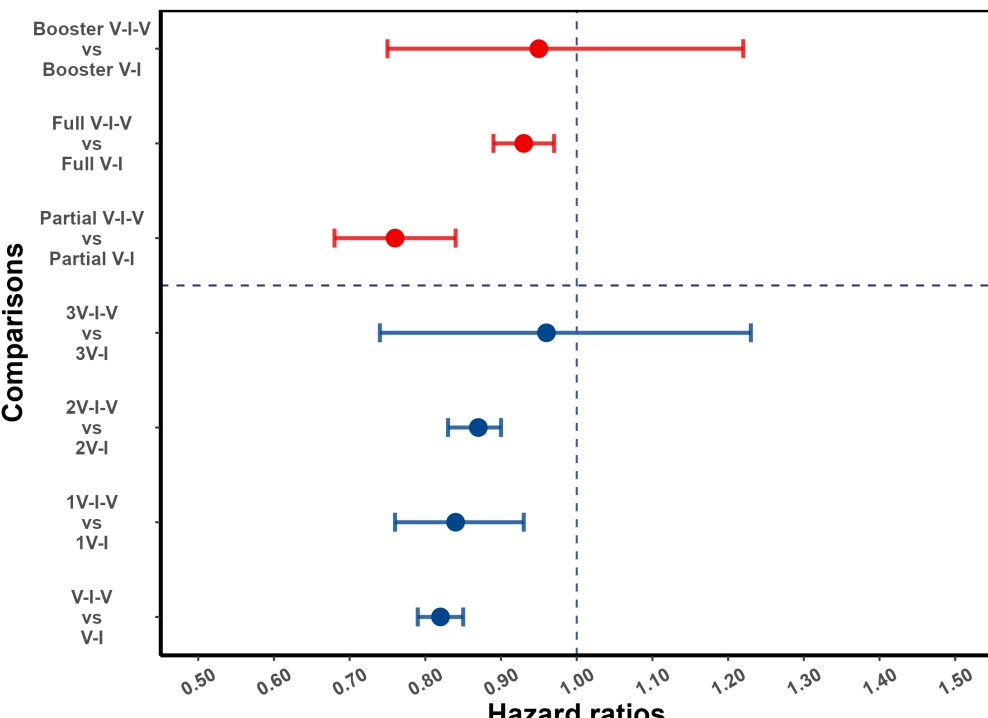

**Appendix 1—figure 2.** Effect of post-infection vaccination on severe acute respiratory syndrome coronavirus 2 (SARS-CoV-2) reinfection stratified by pre-infection vaccination and not adjusted for the severity of the first infection. Error bars (95% confidence intervals [CIs]) and dots represent aHR for SARS-CoV-2 reinfection estimated using Cox proportional hazards models. 1V-I-V, 2V-I-V, and 3V-I-V correspond to 1, 2, and 3 vaccine doses before infection, then vaccination, respectively; they were compared to 1V-I, 2V-I, and 3V-I, respectively. V-I-V, Partial V-I-V, Full V-I-V, and Booster V-I-V represent any pre-infection vaccination, partial vaccination, full vaccination, and booster vaccination before infection, followed by post-infection vaccination, respectively.

