## [Editor Report · eLife assessment]

This **important** work by Zheng and colleagues uses a large cohort database from Shanghai to identify that post-infection vaccination among previously vaccinated individuals provides significant low to moderate protection against re-infection. The evidence supporting the conclusion is **convincing** with some limitations, e.g., lack of symptom severity as an outcome, and no inclusion of time since infection as an independent variable. This study will be of interest to vaccinologists, public health officials and clinicians.

---

## [Referee Report · Reviewer #1 (Public review)]

Summary:

Zheng and colleagues assessed the real world efficacy of SARS-CoV-2 vaccination against re-infection following the large omicron wave in Shanghai in April, 2022. The study was performed among previously vaccinated individuals. The study successfully documents a small but real added protective benefit of re-vaccination, though this diminishes in previously boosted individuals. Unsurprisingly, vaccine preventative efficacy was higher if the vaccine was given in the month before the 2nd large wave in Shanghai. The re-infection rate of 24% suggests that long-term anti-COVID immunity is very difficult to achieve. The conclusions are largely supported by the analyses. These results may be useful for planning the timing of subsequent vaccine rollouts.

Strengths:

The strengths of the study are a very large and unique cohort based on synchronously timed single infection among individuals with well documented vaccine histories. Statistical analyses seem appropriate. As with any cohort study, there are potential confounders and the possibility of misclassification and the authors outline limitations nicely in the discussion.

Weaknesses:

The authors have addressed each of my points thoroughly.

---

## [Referee Report · Reviewer #2 (Public review)]

Summary:

This paper evaluates the effect of COVID-19 booster vaccination on reinfection in Shanghai, China among individuals who received primary COVID-19 vaccination followed by initial infection, during an Omicron wave.

Strengths:

A large database is collated from electronic vaccination and infection records. Nearly 200,000 individuals are included in the analysis and 24% became reinfected.

Weaknesses:

The authors have revised the manuscript and have provided satisfactory responses to my prior comments.

---

## [Author Response]

The following is the authors’ response to the original reviews.

**Public Reviews:**

**Reviewer #1 (Public Review):**
Summary:Zheng and colleagues assessed the real-world efficacy of SARS-CoV-2 vaccination against re-infection following the large omicron wave in Shanghai in April 2022. The study was performed among previously vaccinated individuals. The study successfully documents a small but real added protective benefit of re-vaccination, though this diminishes in previously boosted individuals. Unsurprisingly, vaccine preventative efficacy was higher if the vaccine was given in the month before the 2nd large wave in Shanghai. The re-infection rate of 24% suggests that long-term anti-COVID immunity is very difficult to achieve. The conclusions are largely supported by the analyses. These results may be useful for planning the timing of subsequent vaccine rollouts.Strengths:The strengths of the study are a very large and unique cohort based on synchronously timed single infection among individuals with well-documented vaccine histories. Statistical analyses seem appropriate. As with any cohort study, there are potential confounders and the possibility of misclassification and the authors outline limitations nicely in the discussion.Weaknesses:(1) Partially and fully vaccinated are never defined and it is difficult to understand how this differs from single, and double, booster vaccines. The figures including all of these groups are a bit confusing for this reason.

We agree with the reviewer that the distinction between these groups could have been made clearer. To address this comment, we modified the legend of the figure that presents hazard ratios based on these two categorisations (here, and throughout this document, changes in the text are underlined):

“Figure 3. Effect of post-infection vaccination on SARS-CoV-2 reinfection stratified by pre-infection vaccination. Error bars (95% CIs) and circles represent aHR for SARS-CoV-2 reinfection estimated using Cox proportional hazards models. V-I-V, 1V-I-V, 2V-I-V, 3V-I-V corresponds to any pre-infection vaccination, 1, 2 and 3 vaccine doses before infection, then vaccination, respectively; they were compared to V-I, 1V-I, 2V-I, 3V-I, respectively. Partial V-I-V, Full V-I-V and Booster V-I-V represent partial vaccination, full vaccination and booster vaccination before infection, followed by post-infection vaccination, respectively. The number of doses received by individuals with partial versus full (and full with booster) vaccination depends on the type of SARS-CoV-2 vaccine received; in Table S3 we present a cross-classification of participants in the analytic population by these vaccination-related categorical variables.”

Further, to facilitate visualisation of Figure 3, and emphasize that estimates are presented based on two different ways of categorising vaccination history, we have now included a horizontal line between estimates based on each category.

Table S3 has been included in the Supplementary Appendix:

(2) Figure 3 is a bit challenging to interpret because it is a bit atypical to compare each group to a different baseline (ie 2V-I-V vs 2V-I). I would label the y-axis 2V-I-V vs 2V-I (change all of the labels) to make this easier to understand.

We agree that having the y-axis tick labels describing both groups being compared, rather than only describing the post-infection vaccination group, will help readers to understand this figure. In our response to the previous comment, we presented an updated version of this figure, where this change was also incorporated (see above).

(3) A 15% reduction in infection is quite low. It would be helpful to discuss if any quantitative or qualitative signals suggest at least a reduction in severe outcomes such as death, hospitalization, ER visits, or long COVID. I am not sure that a 15% reduction in cases supports extra vaccination without some other evidence of added benefit.

Unfortunately, data on the clinical severity of diagnosed SARS-CoV-2 infections were not available. Some previous studies on COVID-19 vaccines observed that effectiveness against severe outcomes was similar or higher than that for outcomes that do not imply severe disease (e.g. infection). For example, in a study in Israel comparing four versus three vaccine doses, Magen and colleagues observed that the effectiveness of a fourth dose, relative to three doses, was 52% against infection, 61% against symptomatic COVID-19, and 76% against COVID-19 related death (Magen et al. Fourth Dose of BNT162b2 mRNA Covid-19 Vaccine in a Nationwide Setting. NEJM 2022; see also, for example, Nasreen et al. Effectiveness of COVID-19 vaccines against symptomatic SARS-CoV-2 infection and severe outcomes with variants of concern in Ontario. Nature Microbiology 2022, or Sacco et al. Effectiveness of BNT162b2 vaccine against SARS-CoV-2 infection and severe COVID-19 in children aged 5–11 years in Italy: a retrospective analysis of January–April, 2022. Lancet 2022). However, this pattern of increasing effectiveness with increasing outcome severity was not consistently reported in all studies or settings. We agree that public health officials who will use our results to guide future vaccination policy in China and abroad need to interpret the results in the context of these other outcomes that were not assessed and of those previous studies, that, although performed in different epidemiological settings, suggest that our analysis does not capture all benefits of post-infection vaccine doses.

We have now included the following statements in the *Discussion* section:

“Finally, data on the severity of infections during the second wave were not available, which prevented analyses of clinical outcomes other than infections (e.g. COVID-19-related hospitalization or death). Although some previous studies (Magen et al. Fourth Dose of BNT162b2 mRNA Covid-19 Vaccine in a Nationwide Setting. NEJM 2022; Nasreen et al. Effectiveness of COVID-19 vaccines against symptomatic SARS-CoV-2 infection and severe outcomes with variants of concern in Ontario. Nature Microbiology 2022) estimated similar or higher vaccine effectiveness against severe outcomes compared to outcomes that presumably include both milder and severe presentations, this pattern was not observed in all studies. Epidemiologists and public health officials who will use our results to define vaccination policy should thus take into account the fact that our analysis does not capture all benefits of post-infection vaccinations.”

(4) Why exclude the 74962 unvaccinated from the analysis. it would be interesting to see if getting vaccinated post-infection provides benefits to this group

The reasons why we focused on individuals who had been vaccinated before their first infection were two: (i) in most settings, including those with SARS-CoV-2 epidemiologic history similar to that of Shanghai, a high percentage of the population has received vaccine doses; (ii) in settings with high vaccination coverage, the group of individuals who remain unvaccinated despite widespread availability of vaccines likely differs from those who have been vaccinated – for example, with regard to behavioural factors and comorbidity profile. Having said that, we agree that reporting analyses for the group of individuals who had not been vaccinated before first infection might be informative. We have thus included in the Supplementary Appendix a short section that reports results for this group of patients; Table S4 also presents these estimates.

“Effect of post-infection vaccination in individuals with no history of vaccination before infection

In this supplementary section, we present findings for individuals who were unvaccinated before infection during the first Omicron variant wave in Shanghai. For this group of individuals, post-infection vaccination did not confer significant protection against reinfection (adjusted hazard ratio [aHR] 1.06, 95% CI 0.97, 1.16). The analysis indicates that the effect of post-infection vaccine doses was not significant in both female (aHR 0.97 [0.84, 1.11]) and male individuals (aHR 1.12 [0.99, 1.26]), as well as for participants aged 60 years or older (aHR 0.92 [0.82, 1.04]) and younger adults (20-60 years) (aHR 1.12 [0.92, 1.37]). These results suggest that, in the context of the two Omicron variant waves in Shanghai, a first vaccine dose administered after infection did not provide a clear benefit in terms of reducing risk of subsequent infections for those not previously vaccinated.”

We refer to this new analysis in the Results section:

“For individuals who had received at least one vaccine dose before infection during the first Omicron variant wave, post-infection vaccination was protective against reinfection (adjusted hazard ratio [aHR] 0.82, 95% CI 0.79, 0.85). As shown in Figure 3, this protective effect was observed in subgroups defined by the number of pre-infection vaccine doses: aHR of 0.84 (95% CI, 0.76, 0.93) and 0.87 (95% CI, 0.83, 0.90) for one and two pre-infection doses respectively; and for patients with three vaccine doses prior to infection, the association was not statistically significant (aHR: 0.96 [0.74, 1.23]). When analyses are stratified by partial and full vaccination status before the first infection, an additional vaccine dose was protective (aHR 0.76 [0.68, 0.84], and 0.93 [0.89, 0.97], respectively); and among individuals who had received booster vaccination before the spread of the first Omicron variant wave in Shanghai, the hazard ratio estimate was consistent with a more limited effect (aHR: 0.95 [0.75, 1.22]). For comparison, results for individuals who had not been vaccinated before their first infection are shown in the Supplementary Appendix (supplementary section “Effect of post-infection vaccination in individuals with no history of vaccination before infection” and Table S4)”

(5) Pudong should be defined for those who do not live in China.

We have now included a sentence defining Pudong in the *Methods* section:

“This study included individuals diagnosed with their first SARS-CoV-2 infection between April 1 and May 31, 2022 in the Pudong District, which is a large and densely populated district of Shanghai spanning an area of 1,210 square kilometers with a permanent resident population of 5.57 million, served by more than 30 hospitals and 60 community health centers;… ”

(6) The discussion about healthcare utilization bias is welcomed and well done. It would be great to speculate on whether this bias might favor the null or alternative hypothesis.

We believe the reviewer is referring to the following statement:

“Differences in healthcare-seeking behavior could also bias case ascertainment between post-infection vaccinated and unvaccinated individuals, although, as we restricted the study population to individuals who had received at least one pre-infection dose, this potential bias might be more limited than in other vaccine studies.”

Bias linked to healthcare seeking behaviour could affect the association between vaccination and infection in two different ways: individuals who are more health conscious are more likely to get vaccinated and also to seek medical care when infected, and this would bias results toward null; however, if the same individuals are also more likely to avoid exposure to potentially infectious individuals, their behaviour could also bias results in the opposite direction – that is, it would appear to increase vaccine effectiveness. As mentioned in the Discussion section, we expected this bias to be limited. We have now modified the paragraph:

“Differences in healthcare-seeking behavior could also bias case ascertainment between post-infection vaccinated and unvaccinated individuals. Although we restricted the study population to individuals who had received at least one pre-infection vaccination, which suggests a higher degree of homogeneity in healthcare-seeking behaviour compared to that in the total population, it is possible that this bias might have affected our estimates. For example: individuals who were more health conscious might have been more likely to receive post-infection vaccination and also more likely to seek medical care or testing when reinfected, and this would have biased results toward the null; it is, however, also conceivable that these individuals were more likely to avoid contact with potentially infectious persons, which could have biased results in the opposite direction.”

**Reviewer #2 (Public Review):**
Summary:This paper evaluates the effect of COVID-19 booster vaccination on reinfection in Shanghai, China among individuals who received primary COVID-19 vaccination followed by initial infection, during an Omicron wave.Strengths:A large database is collated from electronic vaccination and infection records. Nearly 200,000 individuals are included in the analysis and 24% became reinfected.Weaknesses:The article is difficult to follow in terms of the objectives and individuals included in various analyses. There appear to be important gaps in the analysis. The electronic data are limited in their ability to draw causal conclusions.More detailed comments:In multiple places (abstract, introduction), the authors frame the work in terms of understanding the benefit of booster vaccination among individuals with hybrid immunity (vaccination + infection). However, their analysis population does not completely align with this framing. As best as I can tell, only individuals who first received COVID-19 vaccination, and subsequently experienced infection, were included. Why the analysis does not also consider individuals who were infected and then vaccinated is not clear.

The focus of our analysis is on the most frequent scenario in many countries: settings where a high proportion of the population has been vaccinated. As mentioned in our response to a comment from Reviewer #1, those individuals who remain unvaccinated after the first years of this pandemic are likely to be different, with respect to many factors, from individuals with history of SARS-CoV-2 vaccination. Further, differences between unvaccinated and vaccinated individuals are likely setting-specific, linked to local availability of and access to vaccination, cultural differences in healthcare seeking behaviour, and possible differences in the frequencies of medical conditions that might influence (promote or prevent) vaccine uptake. We prefer to keep the focus of this work on individuals who had been vaccinated before their first infection; however, we have now included in the Supplementary Appendix a section, presented in a response to Reviewer #1, that reports results for this group of individuals.

In vaccine effectiveness analyses, why was time since initial infection not examined as a modifier of the booster effect? Time since the onset of the Omicron wave is only loosely tied to the immune status of the individual.

We agree with the reviewer that assessing effect modification by the time since initial infection would be important. However, in Shanghai, most initial infections occurred during a narrow time window relative to the time window between the first and second Omicron variant waves. Indeed, as mentioned in the *Results* section, most first infections (243,906, 88.8%) occurred in April; for 306 (0.1%) individuals, information on the date of first infection was not available. Given this narrow time window and in order to limit the number of comparisons in our study, we preferred not to investigate this aspect of the hybrid immunity. In settings where multiple SARS-CoV-2 waves occurred, over a longer period of time, which would imply sufficient variation in this variable “time since initial infection”, we believe that it would be essential to account for this.

The effect of booster vaccination on preventing symptomatic vs. asymptomatic reinfection does not appear to have been evaluated; this is a key gap in the analysis and it would seem the data would support it.

Not having clinical presentation data is a limitation in our study. That is a weakness of many real-world vaccine effectiveness analyses based large medical and administrative datasets. We have now explicitly mentioned this in the Discussion section.

“Finally, data on the severity of infections during the second wave were not available, which prevented analyses of clinical outcomes other than infections (e.g. COVID-19-related hospitalization or death). Although some previous studies (Magen et al. Fourth Dose of BNT162b2 mRNA Covid-19 Vaccine in a Nationwide Setting. NEJM 2022; Nasreen et al. Effectiveness of COVID-19 vaccines against symptomatic SARS-CoV-2 infection and severe outcomes with variants of concern in Ontario. Nature Microbiology 2022) estimated similar or higher vaccine effectiveness against severe outcomes compared to outcomes that presumably include both milder and severe presentations, this pattern was not observed in all studies. Epidemiologists and public health officials who will use our results to define vaccination policy should thus take into account the fact that our analysis does not capture all benefits of post-infection vaccinations.”

In lines 105-108, the demographic description of the analysis population is incomplete. Is sex or gender identity being described? Are any individuals non-binary? What is the age distribution? (Only the proportions 20-39 and under 6 are stated.)

We have now clarified in the manuscript that only information on sex at birth was provided by the Center for Disease Control and Prevention in Shanghai. We made the following change in the Methods section:

“Information on infection history as well as data on demographic variables (sex at birth, and age) were provided by Center for Disease Control and Prevention in Shanghai, China”

We have also modified the legend of Table 1:

“Table 1. Characteristics of the study population and reinfection rate by post-infection vaccination status. Here, reinfection rate refers to the percentage of the relevant study subpopulation with evidence of reinfection between December 1, 2022 and January 3, 2023. Note that for the variables on region, occupation, and clinical severity, data are missing for large fractions of the study population. Note also that information was only available on sex at birth, but not on gender.”

Regarding the reviewer’s comment on the age distribution, this information is presented for the following categories in Table 1: 0-6 years, 7-19 years, 20-39 years, 40-59 years, and 60+ years. However, we had not referred to Table 1 in the section 3.1 of the manuscript. We have now corrected that:

“To assess the effect of an additional vaccine dose given after infection, the analytic sample consisted of 199,312 individuals (Figure 1). 85,804 were women (43.1%); 836 (0.4%) had gender information missing. 38.1% of the study participants were aged 20 to 39 years and only 0.9% were aged 0 to 6 years (see Table 1 for additional information).”

Figure 1 consort diagram is confusing. In the last row, are the two boxes independent or overlapping sets of individuals? Are all included in secondary analyses?

We agree that additional information should have been provided in the legend. The boxes represent overlapping sets of individuals – that is, some individuals were included in both secondary analyses in the box on the left and in the box on the right. These analyses involved different ways of categorizing individuals. Below is the updated figure legend:

“Figure 1. Flow chart describing the selection of participants for the analysis. The number of individuals in this figure is not the same as some of the numbers in Table 1 because of missing data in key variables. Note that in the bottom part of the chart, related to secondary analyses, the boxes represent overlapping sets of study participants; in other words, some individuals included in the secondary analyses that correspond to the left box were also included in analyses corresponding to the box on the right.”

**Recommendations for the authors:**

**Reviewer #2 (Recommendations For The Authors):**
Minor comment: the terms "vaccination"/"vaccinated" are used both to refer to the primary vaccination (pre-initial infection) and to the booster vaccination (post-initial vaccination), and this causes confusion.

Thank you. We have now revised the manuscript (Methods, Results and Discussion sections) to use the terms “post-infection vaccination” and “post-infection vaccinated” to reduce ambiguity. We also included the following statement in the Background section:

“In December 2022, an important change in the COVID-19 policy in China, namely the end of most social distancing measures and of mass screening activities, was associated with a second surge in SARS-CoV-2 infections in Shanghai. The current circulation of the virus in the Shanghainese population and reports of vaccine fatigue mean that it is important to estimate the protective effect of vaccination against reinfection in this population. In this study, we aimed to quantify the effect of vaccine doses given after a first infection on the risk of subsequent infection. For that, we used data collected during the first Omicron variant wave, when hundreds of thousands of individuals tested real-time polymerase chain reaction (RT-PCR)-positive for SARS-CoV-2 infection8 in Shanghai, of which 275,896 individuals in Pudong. The fact that the population in Shanghai was mostly SARS-CoV-2 infection naïve before the spread of the Omicron variant provides a unique opportunity to estimate the real-world benefit of post-infection vaccine doses in a population that was first exposed to infection during a relatively short and well-defined time window. We further investigated whether the number of pre-infection vaccination doses modified the protective effect of the post-infection dose against Omicron BA.5 sublineage. To avoid ambiguity in the text, in the following sections, we often refer to vaccine doses given after the initial infection as “post-infection vaccination” or “post-infection vaccine doses”.